# Put CASH on Bandits: A Max K-Armed Problem for Automated Machine Learning

**Amir Rezaei Balef, Claire Vernade and Katharina Eggensperger**
Department of Computer Science, University of Tübingen
{amir.rezaei-balef, claire.vernade, katharina.eggensperger}@uni-tuebingen.de

## Abstract

The Combined Algorithm Selection and Hyperparameter optimization (CASH) is a challenging resource allocation problem in the field of AutoML. We propose MaxUCB, a max $k$-armed bandit method to trade off exploring different model classes and conducting hyperparameter optimization. MaxUCB is specifically designed for the light-tailed and bounded reward distributions arising in this setting and, thus, provides an efficient alternative compared to classic max $k$-armed bandit methods assuming heavy-tailed reward distributions. We theoretically and empirically evaluate our method on four standard AutoML benchmarks demonstrating superior performance over prior approaches. We make our code and data available at https://github.com/amirbalef/CASH_with_Bandits.

## 1 Introduction

The performance of machine learning (ML) solutions is highly sensitive to the choice of algorithms and their hyperparameter configurations which can make finding an effective solution a challenging task. AutoML aims to reduce this complexity and make ML more accessible by automating these critical choices [Hutter et al., 2019, Baratchi et al., 2024].

For example, Hyperparameter optimization (HPO) methods focus on finding well-performing hyperparameter settings given a resource constraint, such as an iteration count or a time limit. However, in practice, it is often unclear which ML model class would perform best on a given dataset [Bischl et al., 2025]. The problem of jointly searching the model class and the appropriate hyperparameters has been coined CASH, Combined Algorithm Selection and Hyperparameter optimization [Thornton et al., 2013]. As a prime example, on tabular data, a ubiquitous data modality [van Breugel and van der Schaar, 2024], the state-of-the-art ML landscape covers classic ML methods, ensembles of gradient-boosted decision trees and modern deep learning approaches [Kadra et al., 2021, Gorishniy et al., 2021, 2024, McElfresh et al., 2023, Kohli et al., 2024, Hollmann et al., 2023, Holzmüller et al., 2024].

A popular approach to address the CASH problem is to use categorical and conditional hyperparameters to run HPO directly on the combined hierarchical search space of models and hyperparameters. AutoML systems use this approach, which we call *combined search*, to search well-performing ML pipelines [Thornton et al., 2013, Feurer et al., 2015, Komer et al., 2014, Kotthoff et al., 2017, Feurer et al., 2022], but HPO remains inefficient in high-dimensional and hierarchical search spaces. A naive solution to address the scalability limitation is to run HPO independently for the smaller search spaces of each ML model class and then compare the found solutions. However, this solution often exceeds available computational resources and does not scale well with an increasing number of ML models. Figure 1 (left) illustrates the difference between searching each space individually (colored dashed lines) and *combined search* (black line) on an exemplary dataset.

39th Conference on Neural Information Processing Systems (NeurIPS 2025).

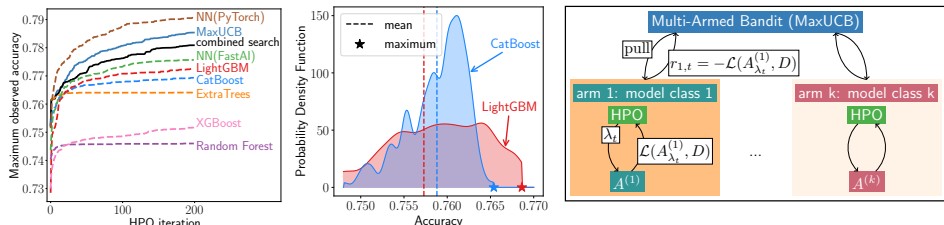

Figure 1: (Left) MaxUCB (blue line) *outperforms combined search* (black line) to identify the best-performing model class (brown line). (Middle) The irregular distribution of the empirical performance of model classes is left-skewed, and *a higher mean may not correspond to a higher maximum*. (Right) MaxUCB selects for which model to run one iteration of HPO during *two-level optimization*.

To leverage the efficiency of HPO in low dimensions, we use a Multi-Armed Bandit (MAB) method [Lattimore and Szepesvári, 2020] to dynamically allocate our budget. As shown in Figure 1 (right), each time the bandit strategy pulls an arm, it runs one iteration of HPO to evaluate a new configuration, resulting in a loss. The negative of the loss is used as reward feedback for the bandit algorithm. This approach is known as *two-level CASH (decomposed CASH)* [Hoffman et al., 2014, Liu et al., 2019].

While most classical MAB problems aim to maximize the average rewards over time, the goal of the bandit algorithm for *two-level CASH* should be to maximize the maximum reward observed over time: as illustrated in Figure 1 (middle), tuning the LightGBM (red) will eventually outperform CatBoost. This goal aligns with Max $K$-Armed Bandit (MKB) problems [Carpentier and Valko, 2014, Achab et al., 2017, Baudry et al., 2022], often referred to as Extreme Bandits.

In the context of AutoML, the time horizon is limited, making efficiency crucial. Prior work on *two-level CASH* found that state-of-the-art MKB algorithms are not sufficiently sample-efficient in practice [Hu et al., 2021, Balef et al., 2024]. Additionally, Nishihara et al. [2016] argued that the parametric assumptions derived from extreme value theory are not applicable in the context of HPO and stated that determining "what realistic assumptions are likely to hold in practice for hyperparameter optimization is an important question".

We precisely address this open question through a thorough statistical analysis of the empirical reward distributions of HPO tasks. **Our main contribution is a state-of-the-art algorithm for decomposed CASH based on a novel extreme bandit algorithm we call MaxUCB** (Algorithm 1). We demonstrate the performance of our method on four benchmarks, which highlights the relevance of our assumptions for a wide variety of CASH problems. We analyze the theoretical performance of MaxUCB (Theorem 4.2) through regret bounds that also justify our novel choice of exploration bonus for the type of distributions relevant to the CASH problem. Importantly, our objective is rather to propose a practical algorithm with good empirical performance on CASH rather than a general multi-purpose bandit algorithm, so our guarantees hold under carefully crafted assumptions that resolve previously open questions [Nishihara et al., 2016].

## 2 Solving CASH using Bandits

The CASH problem for supervised learning tasks is defined as follows [Thornton et al., 2013]. Given a dataset $\mathbb{D} = \{D_{train}, D_{valid}\}$ of a supervised learning task, let $\mathcal{A} = \{A^{(1)}, ..., A^{(K)}\}$ be the set of $K$ candidate ML algorithms, where each algorithm $A^{(i)}$ has its own hyperparameter search space $\mathbf{\Lambda}^{(i)}$. The goal is to search the joint algorithm and hyperparameter configuration space to find the optimal algorithm $A^{(i^*)}$ and its optimal hyperparameter configuration $\boldsymbol{\lambda}^*$ that minimizes a loss metric $\mathcal{L}$, e.g., the validation error[1]. Formally,

$$A^{(i^*)}_{\boldsymbol{\lambda}^*} \in \underset{A^{(i)} \in \mathcal{A}, \boldsymbol{\lambda} \in \mathbf{\Lambda}^{(i)}}{\arg\min} \mathcal{L}(A^{(i)}_{\boldsymbol{\lambda}}, \mathbb{D}). \tag{1}$$

---

[1]We note that $\mathcal{L}$ can also be the result of k-fold cross-validation or other evaluation protocols measuring the expected performance of a model on unseen data [Raschka, 2018].

For our approach, we study the decomposed variant [Hoffman et al., 2014, Liu et al., 2019] and address the following two-level optimization problem depicted in Figure 1 (right): at the upper level, we aim to find the overall best-performing ML model $A^{(i^*)}$ by selecting model $A^{(i)} \in \mathcal{A}$ iteratively, and at the lower level, we aim to find the best-performing configuration $\boldsymbol{\lambda^*} \in \boldsymbol{\Lambda}^{(i)}$ for the selected model $A^{(i)}$. Formally,

$$A^{(i^*)} \in \arg\min_{A^{(i)} \in \mathcal{A}} \mathcal{L}(A^{(i)}_{\boldsymbol{\lambda^*}}, \mathbb{D}), \quad \text{s.t.} \quad \boldsymbol{\lambda}^* \in \arg\min_{\boldsymbol{\lambda} \in \boldsymbol{\Lambda^{(i)}}} \mathcal{L}(A^{(i)}_{\boldsymbol{\lambda}}, \mathbb{D}). \tag{2}$$

The right-hand side of Equation 2 in the lower level can be efficiently addressed by existing iterative HPO methods such as Bayesian optimization (BO) [Jones et al., 1998, Garnett, 2022], which has been demonstrated to perform well in practical settings [Snoek et al., 2012, Chen et al., 2018, Cowen-Rivers et al., 2022]. BO fits a surrogate model and uses an acquisition function to find a promising configuration to evaluate next. On the upper level, or left-hand side of Equation 2, the challenge is to carefully allocate the budget $T$ of HPO runs to the $K$ models in a manner that trades off exploration of the hyperparameter space of all models and exploitation (optimization) of the most promising model. As already noted in previous work, this is a typical MAB problem [Cicirello and Smith, 2005, Streeter and Smith, 2006a, Nishihara et al., 2016, Metelli et al., 2022].

At time $t$, the bandit algorithm chooses model $I_t \in \mathcal{A}$, and we denote $\boldsymbol{\lambda}_t$ the configuration proposed by the HPO method in the lower level. As a reward $r_{i,t}$, we feed back to the bandit algorithm an evaluation of the negative loss:

$$r_{i,t} = -\mathcal{L}(A^{(i)}_{\boldsymbol{\lambda}_t}, \mathbb{D}). \tag{3}$$

In general, and as opposed to standard MAB, this reward process is not i.i.d. conditionally on the arm choices because the loss of the models depend on the progress of HPO on each model class (arm) as well as additional loss evaluation noise. To be able to design a tractable bandit algorithm, it is crucial to find an appropriate way to model this process to build controllable estimators. We focus on this aspect in the next section.

To complete the bandit model of this problem, we need to choose a regret metric that defines the oracle objective we compare to, and indeed aligns with Equation 2. For HPO, the regret should target max-value objectives [Jamieson and Talwalkar, 2016, Nishihara et al., 2016]. Therefore, in this work, we propose to minimize the (extreme) regret $R(T)$:

$$R(T) = \max_{k \le K} \mathbb{E}[\max_{t \le T} r_{k,t}] - \mathbb{E}[\max_{t \le T} r_{I_t,t}], \tag{4}$$

where the expectation is over the stochasticity of both the HPO procedure (e.g., random search or Bayesian optimization), the ML models themselves (e.g., random initialization and training variability), and the loss evaluation procedure at each round. The regret measures the gap between the expected performance of the bandit algorithm (right part) and that of the oracle model that would achieve the lowest loss, should we assign the full budget of HPO runs to it (left part). So this objective is indeed aligned with Equation 2 and fully integrates the budget constraints.

Instead of using MKB algorithms to directly address Equation 4, related prior works focus on alternative methods. For example, Hu et al. [2021] proposed and analyzed the *Extreme-Region Upper Confidence Bounds (ER-UCB)* algorithm maximizing the extreme region of the feedback distribution, assuming Gaussian rewards. More recently, Balef et al. [2024] have shown that existing MKB algorithms underperform when applied to *two-level CASH*, and they proposed methods for maximizing the quantile values instead of the maximum value.

As another alternative method, Li et al. [2020] framed the CASH problem as a Best Arm Identification (BAI) task and introduced the *Rising Bandits* algorithm [Li et al., 2020]. This MAB method assumes that the reward function for each arm increases with each pull, following a rested bandit model with non-decreasing payoffs [Heidari et al., 2016] (which has been shown to have linear regret when the reward increment per pull exceeds a threshold [Metelli et al., 2022]). *Rising Bandits* can be used for our setting using the maximum observed performance of the HPO history as the reward. However, this algorithm assumes deterministic rewards and increasing concave reward functions. To weaken this assumption, Li et al. [2020] introduced a hyperparameter to increase initial exploration. Mussi et al. [2024] further weakened this assumption by assuming that the moving average of the rewards is an increasing concave function.

Generally, in BAI approaches, the goal is to identify the arm with the highest mean reward. Here, rewards are the result of HPO runs, so this objective does not align well with Equation 2: there is no reason to measure the quality of a model on average over a random subset of hyperparameters chosen by HPO. This approximation made by prior work is justified by the complexity of this modeling problem and the existence of solid foundations on BAI to build upon. But in this work, we propose a fully data-driven model that fits better the true CASH objective and results in better empirical results.

## 3   Data Analysis of HPO Tasks

The reward process in Equation 3 is complex as it depends on the model and on the chosen HPO algorithm. So, as discussed, it is necessary to model it. Rather than choosing a convenient parametric family, we conduct a thorough analysis of typical sequences of losses obtained on real benchmarks. For each ML model class (corresponding to arms in our setup), we run $T = 200$ iterations of HPO, with 32 repetitions (each using a different seed) on a varying number of datasets on four AutoML benchmarks (see Appendix D.2 and C.1).

We first analyze the *survival function* of the reward distributions. Recall that for a random variable $X \sim d$, the survival function is defined by: $x \mapsto G(x) = P_{X \sim d}(X \geq x)$. Figure 2 shows the average empirical survival function of all observed performances (normalized between 0 and 1 for each task) for each arm. We rank model classes (i.e., arms) based on their best performance per dataset and report results on two benchmarks (see Appendix C for more details and results on all benchmarks). We observe that the reward distributions are

1. **bounded**. The reward, which measures the performance of a model, is determined by a score metric, e.g., accuracy. The extreme values vary for each arm and depend on the capability of the model class and the complexity of the task. Therefore, even if we run HPO indefinitely, achieving an infinite reward is impossible. Consequently, each arm has a bounded support with different maximum values, and at least a single optimal arm exists. $\rightarrow$ **We can define a sub-optimality gap (Definition 3.1).**

2. **short-tailed, left-skewed, and the right tail is not heavy**. The rewards are concentrated near the maximal value per model class, and extreme events are not outliers. As the HPO method's performance reaches a certain level, further optimization often yields only small gains as optima tend to have flat regions [Pushak and Hoos, 2022]. Therefore, many configurations perform similarly well, resulting in a skewed distribution.[2] $\rightarrow$ **We expect to observe many extreme rewards.**

3. **nearly stationary**. This means that the optimal arm does not change over time. This is clear for TabRepoRaw. We observe significant changes in the distributions for YaHPOGym, but most of the sub-optimal arms remain sub-optimal over time. $\rightarrow$ **Ranking of well-performing arms does not change over time.**

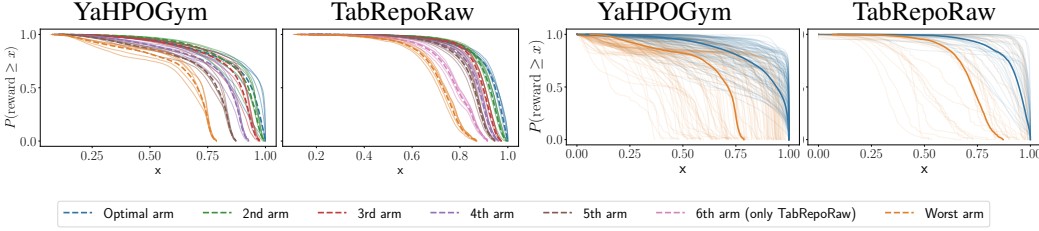

Figure 2: (Left) The average empirical survival function of the reward distribution per arm ranked per dataset. Thin lines correspond to segments of the reward sequence and show the distribution change over time. (Right) The average empirical survival function per dataset for the best and worst arm. Thin lines correspond to individual datasets.

Prior research has shown that existing MKB algorithms can underperform if their assumptions about the distributions are too weak or misaligned [Nishihara et al., 2016, Balef et al., 2024]. This

---

[2]This has also been observed for related tasks, e.g., neural architecture search on CIFAR-10 [Su et al., 2021].

underperformance is largely due to our second observation, which significantly differs from the common assumptions of the existing algorithms.

**Preliminaries.** Based on these observations, we can formulate the following definition and assumption on which we develop and analyze our algorithm.

**Definition 3.1.** The *suboptimality gap* $\Delta_i \geq 0$ for arm $i$ is defined as:

$$\Delta_i = \mathbb{E}\left[\max_{t \leq T} r_{i^*,t}\right] - \mathbb{E}\left[\max_{t \leq T} r_{i,t}\right]$$

where $r_{i^*,t}$ and $r_{i,t}$ are the rewards observed from optimal arm $i^*$ and arm $i$ at time $t$, respectively.

**Assumption 3.2.** We assume that the i.i.d. random variable $X$, representing the rewards, follows a bounded distribution with support in $[a, b]$ and continuous survival function $G$.

**Lemma 3.3.** *Suppose Assumption 3.2 holds. Then, there exists $L, U \geq 0$ such that the survival function $G$ can be bounded near $b$ by two linear functions:*

$$\forall \epsilon \in (0, b - a), \quad L\epsilon \leq G(b - \epsilon) \leq U\epsilon. \tag{5}$$

Lemma 3.3 (proof in Appendix B.1) provides a way to characterize the shape of any bounded distribution near its maximum value through distribution-dependent constants, $L$ and $U$ (see Appendix C.2 for more details and a visualization). Intuitively, it quantifies the behavior of the distribution of the ML model performance in a given hyperparameter space near the optimum. A large value of $L$ indicates a steep drop in the survival function near the maximum, while a small value of $U$ leads to a more gradual decay of the survival function and conversely (see Figure 3).

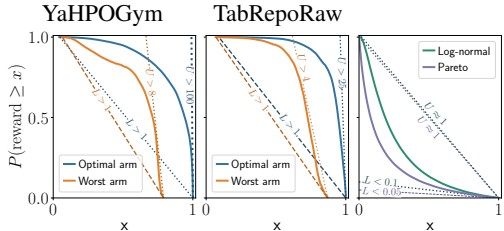

Figure 3: (Left, Middle) L and U for the average survival functions in Figure 2. (Right) We highlight the difference to right-skewed Log-normal and Pareto distributions.

Figure 4: Histogram of $L$ and $U$ values across individual datasets.

We study the values of $L$ and $U$ in Lemma 3.3 on our empirical data and show the distribution of $L$ and $U$ in Figure 4 (see Appendix C.3 for more analyses). Next, we focus on the uniqueness of our assumptions before discussing how this impacts our MKB algorithm's performance in Section 4.

**Common Max $K$-armed Bandit assumptions are misaligned with CASH.** Existing MKB algorithms can be classified into two main categories. First, distribution-free approaches do not leverage any assumption on the type of reward distributions [Streeter and Smith, 2006b, Bhatt et al., 2022, Baudry et al., 2022]. Secondly, parametric and semi-parametric approaches typically assume that the reward distributions follow heavy-tailed distributions, typically following a second-order Pareto assumption whose parameters can be estimated using extreme value theory [Carpentier and Valko, 2014, Achab et al., 2017].

Our empirical analysis shows that existing MKB algorithms' assumptions about reward distributions are either overly broad or misaligned with the CASH problem. We identify two key differences between our assumptions and those made by current MKB algorithms. First, the sub-optimality gap makes the regret definition (Equation 4) for bounded distributions meaningful. Without the existence of a nonzero gap, the regret definition fails since any policy consistently selecting a single arm can achieve zero regret, as Nishihara et al. [2016] pointed out with Bernoulli distributions. Second, our reward distributions differ from those used in existing MKB algorithms. Lemma 3.3 characterizes the shape of the distribution, with a higher $L$ ensuring more mass near extreme values, making extreme values easier to estimate. In our case, values for $L$ are mostly higher than 1 while for heavy-tailed distributions, commonly used as the basis for MKB algorithms, are close to 0 (see the rightmost plot

in Figure 3). In general, our analysis shows that considering the right range of $L$ values unlocks the problem raised by Nishihara et al. [2016], who focused on constructing counterexamples through the distributions that have unrealistic values for $L$. Thus, under our assumptions, their negative result[3] does not apply here.

## 4  MaxUCB

Based on Lemma 3.3 and the regret definition (Equation 4), we introduce MaxUCB in Algorithm 1 for $K$ arms with a limited time horizon of $T$ iterations.[4]

**Description of MaxUCB.** Our algorithm balances exploration and exploitation according to the standard optimism principle at the heart of Upper Confidence Bound (UCB) bandit methods [Auer, 2002, Lattimore and Szepesvári, 2020]. The main novelty we introduce is in the computation of a distribution-adapted exploration bonus for MaxUCB.

Our exploration bonus $(\frac{\alpha \log(t)}{n})^2$ deviates from typical UCB literature due to faster concentration of maximum values in bounded distributions. This is because the probability of bad events (violating confidence intervals for the expectation of the max, see Equation 30 in Appendix B.3) can be written as:

$$P(\text{Bad events}) \leq O\left(e^{-n\sqrt{C(n)}} + nC(n)\right), \tag{6}$$

where distribution-dependent constants are hidden for clarity. Then, setting $C(n) = 1/n^2$ minimizes the probability of bad events: the first term becomes independent of $n$, while the second term decreases with $n$. Notably, this faster concentration can only be obtained for the reasonably well-behaved distributions we consider following the study of the previous section and it is not a general property of the maxima; more details can be found in Appendix B.4. Furthermore, MaxUCB uses $\alpha \geq 0$ to control the exploration-exploitation trade-off; a higher $\alpha$ leads to more exploration.

---

**Algorithm 1** MaxUCB

**Require:**  $\alpha$ (exploration parameter) , $T$ (time horizon), $K$ (arms)
1: **for** each arm $i \leq K$ **do**
2:     Pull arm $i$, set $n_i \leftarrow 1$, observe reward $r_{i,1}$                    ▷ Evaluating default configuration
3: **end for**
4: **for** $t = K + 1$ to $T$ **do**
5:     **for** each arm $i \leq K$ **do**
6:         Update policy $U_i = \max{(r_{i,1}, ..., r_{i,n_i})} + (\frac{\alpha \log(t)}{n_i})^2$   ▷ differs from classical UCB, where $U_i = \bar{r}_i + \sqrt{\frac{\alpha \log(t)}{n_i}}$
7:     **end for**
8:     Select arm $I_t = \underset{i \leq K}{\arg\max}\, U_i$ ,    $n_{I_t} \leftarrow n_{I_t} + 1$ , then observe reward $r_{I_t, n_{I_t}}$
9: **end for**

---

**Analysis of MaxUCB.** We first show a regret decomposition result specific to max $K$-armed bandits that directly relates the regret definition in equation 4 with the number of suboptimal trials.

**Proposition 4.1.** *(Regret Upper Bound) the regret upper bound up to time $T$ is related to the number of times sub-optimal arms are pulled:*

$$R(T) \leq \frac{\max\limits_{i \leq K} b_i}{T} \sum_{i \neq i^*}^{K} N_i(T) \tag{7}$$

*where $N_i(T) = \mathbb{E}(\sum_{t=1}^{T} \mathbb{1}\{I_t = i\})$ is the number of sub-optimal pulls of arm $i$, and $b_i$ is the upper bound on the support of the rewards of arm $i$.*

---

[3]Theorem 11 in [Nishihara et al., 2016]: "no policy can be guaranteed to perform asymptotically as well as an oracle that plays the single best arm for a given time horizon.", which means any policy needs to explore all arms for budget T

[4]MaxUCB needs to store the number of pulls and the maximum reward for each arm, resulting in a memory requirement of $\mathcal{O}(K)$. The time complexity is $\mathcal{O}(KT)$.

The proof is provided in Appendix B.2 and relies on standard tools in the extreme bandit literature [Baudry et al., 2022]. From this result, it is clear that we can now obtain an upper bound on the regret by controlling the number of suboptimal arm pulls $(N_i(T))_{i \neq i^*}$ individually. Our main theoretical result below proves such an upper bound for Algorithm 1.

**Theorem 4.2.** *For any suboptimal arm $i \neq i^*$, the number of suboptimal draws $N_i(T)$ performed by MaxUCB (**Algorithm 1**) up to time $T$ is bounded by*

$$N_i(T) \leq \frac{T^{1-2L_{i^*}\alpha\sqrt{\Delta_i}}}{1 - 2L_{i^*}\alpha\sqrt{\Delta_i}} + 2\alpha\sqrt{U_i T}\log(T). \tag{8}$$

The result of Theorem 4.2 (proof in Appendix B.3) highlights that MaxUCB primarily leverages two key properties: the sub-optimality gap $\Delta_i$ and the shape of the distribution, as defined in Lemma 3.3. Specifically, the performance improves with a larger sub-optimality gap $\Delta_i$ and higher values of $L_{i^*}$ ($L$ for the optimal arm), which means that samples drawn from the distribution of the optimal arm are likely to be close to the extreme values. Additionally, smaller values of $U_i$ ($U$ for a sub-optimal arm $i$), which means it is less likely to draw samples close to the extreme values, reduce the number selecting sub-optimal arm $i$, thus enhancing overall performance. For our task, as is shown in Figure 3, the values for $L_{i^*}$ are higher than 1 and $U_i$ less than 10 in most cases, yielding high empirical performance. We compare the number of pulls observed in our experiments with the expected values based on our theoretical analysis in Appendix E.2, showing that our analysis is not loose. However, it is essential to note that, in general, finding the optimal arm is challenging if $\Delta_i$ is close to zero or $L_{i^*}$ is very small. We assess the performance of MaxUCB on synthetic tasks in Appendix E.4, showing that the performance of MaxUCB deteriorates on tasks that do not satisfy our assumptions.

Finally, we provide a regret upper bound that combines the decomposition in Proposition 4.1 with the individual upper bounds above. This requires finding a parameter $\alpha$ that resolves a trade-off between the arms and minimizes the total upper bound, as shown in Corollary 4.3, whose proof is immediate.

**Corollary 4.3.** *If $L_{i^*}$, $\min_{i \neq i^*}(\Delta_i)$, and $T$ are known in advance, then the total regret $R(T)$ can be bounded as follows by choosing the exploration parameter $\alpha$ appropriately:*

$$R(T) \leq \mathcal{O}\left(\frac{K\log T}{\sqrt{T}} \max_{i \leq K} b_i\right), \quad \text{when } \alpha = \frac{1}{4L_{i^*}\sqrt{\min_{i \neq i^*}\Delta_i}}\left(1 - \frac{2\log(\log(T))}{\log(T)}\right). \tag{9}$$

This final result helps to understand the role of $\alpha$ and serves as guidelines to choose it in practice. Specifically, Equation 9 shows that either a small value of $L_{i^*}$ or a small sub-optimality gap requires a higher value for $\alpha$. Intuitively, a small $L_i^*$ means that the max value of the best arm needs more samples to be nearly reached, and a small suboptimality gap means that the two best arms are close and so hard to distinguish. Indeed, these problem-dependent quantities are unknown to the practitioner, so a direct approach to calculate $\alpha$ is not feasible. Therefore, evaluating performance robustness under "loose tuning" of $\alpha$ is essential.

## 5 Performance on AutoML tasks

Finally, we examine the empirical performance of MaxUCB in an AutoML setting via reporting average ranking and the number of wins, ties, and losses across tasks for each benchmark (details in Appendix D.1). We first focus on the impact of the hyperparameter $\alpha$ and then compare our approach to others. Specifically, we show that our two-level approach performs better than single-level HPO on the joint space, and MaxUCB outperforms other state-of-the-art bandit methods. To begin, we will provide a brief overview of the experimental setup used across all experiments.

**Experimental setup.** We use four AutoML benchmarks[5], all implementing CASH for tabular supervised learning differing in the considered ML models, HPO method, and datasets, which we

---

[5]We used the AutoML Toolkit (AMLTK) [Bergman et al., 2024]. We ran HPO on a compute cluster equipped with Intel Xeon Gold 6 240 CPUs, requiring 20 000 CPU hours. We conducted the remaining experiments on a local machine with Intel Core i7-1370P, requiring an additional 32 CPU hours.

detail in Table D.1. For TabRepo and Reshuffling, we use available pre-computed HPO trajectories, whereas for TabRepoRaw and YaHPOGym, we use SMAC [Hutter et al., 2011, Lindauer et al., 2022] as Bayesian optimization method to run HPO ourselves.

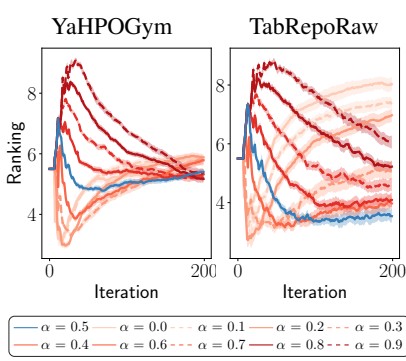

Figure 5: The sensitivity of MaxUCB to hyperparameter $\alpha$, lower is better.

**How sensitive is MaxUCB to the choice of $\alpha$?** Figure 5, shows that the choice of $\alpha$ impacts performance. Lower values of $\alpha$ (light red lines) lead to good performance with small budgets, whereas higher values (dark red lines) achieve stronger final performance when sufficient time is available. An $\alpha$ around $0.5$ yields a balanced trade-off, offering robust anytime performance across tasks (see Appendix D.4 for detailed results). Another insight from this study is that the right choice of $\alpha$ may depend on characteristics of the datasets, such as the support of the losses, and could be meta-learned.

**Competitive Baselines.** We compare against *Quantile Bayes UCB* [Balef et al., 2024], *ER-UCB-S* [Hu et al., 2021] and *Rising Bandits* [Li et al., 2020] which have been developed for the decomposed CASH task. We consider extreme bandits (*QoMax-SDA* [Baudry et al., 2022], *Max-Median* [Bhatt et al., 2022]) and classic *UCB* as general bandit methods. We use default hyperparameter settings for all methods. As *combined search* baselines, we consider Bayesian optimization (*SMAC*) and *random search*.[6] Additionally, we report the performance of the best (oracle) arm.

**How does the two-level approach compare against *combined search*?** We first compare the average rank over time in Figure 6 using *combined search* (black lines; *random search* and *SMAC* if available) to the two-level approach. We observe that all methods outperform *random search* (dotted line) and that while SMAC quickly catches up, some bandit methods continuously achieve a better (lower) rank. Additionally, we observe that most MAB algorithms (except *Rising Bandits* and *QoMax-SDA*) lead to superior performance in the early stages ($T = 50$). This demonstrates that the decomposition is particularly useful when the number of iterations is limited. Additionally, Table 1 shows that the difference in final performance between *combined search* and the two-level approach is significant.

**How does MaxUCB compare against other bandit methods?** Figure 6 shows a substantial difference in the ranking. In the beginning, many methods perform competitively, but MaxUCB yields the best anytime and final performance. Classical *UCB* (red) and extreme bandits (*QoMax-SDA*; brown) perform worst. The *Max-Median* algorithm (purple) shows strong initial performance, but its effectiveness declines with more iterations. While *Max-Median* identifies and avoids the worst arm; it sometimes struggles to select the optimal arm, resulting in non-robust performance.

Next, we look at methods that were originally designed for AutoML. Both *ER-UCB-S* (pink) and *Quantile Bayes UCB* (orange) focus on estimating the higher region of the reward distribution rather than the extreme values. *ER-UCB-S* assumes a Gaussian reward distribution and is consistently outperformed by *Quantile Bayes UCB*, a distribution-free algorithm. *Rising Bandits* (green) underperforms initially due to its costly initialization but reaches a competitive final performance. This is especially pronounced for the YaHPOGym benchmark, where *Rising Bandits* outperforms MaxUCB with respect to normalized average performance (see Figure D.5 in Appendix D.5). This benchmark contains datasets where the optimal arm changes over time. Since *Rising Bandits* models non-stationary rewards, it performs better for these instances.[7]

Finally, MaxUCB and *Quantile Bayes UCB* are the only ones that significantly outperform *combined search* in Table 1. And looking at TabRepoRaw and Reshuffling, as depicted in Figure 6, demonstrates that MaxUCB is a robust method for CASH problems.

---

[6]Only available for TabRepoRaw and YaHPOGym, where we computed HPO trajectories ourselves.

[7]A burn-in phase, i.e., pulling each arm for a few rounds at the beginning without observing the rewards, yields a competitive solution as we assess in Appendix E.3.

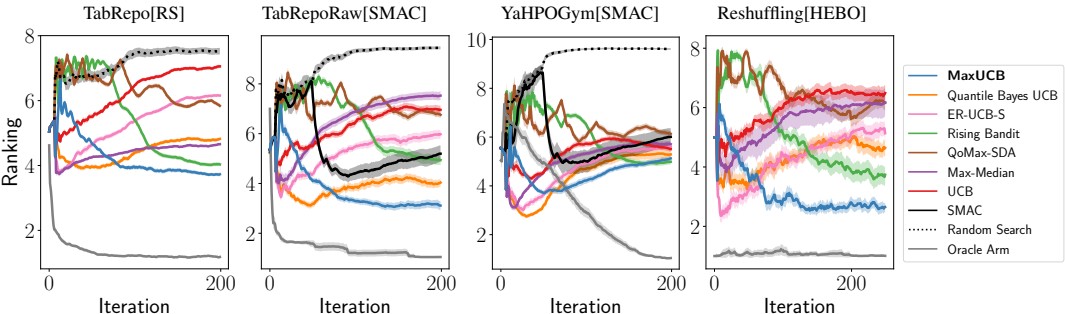

Figure 6: Average rank of algorithms on different benchmarks, lower is better. *SMAC* and *random search* perform *combined search* across the joint space.

| Benchmark | | **MaxUCB** | Quantile Bayes UCB | ER-UCB-S | Rising Bandit | QoMax-SDA | Max-Median | UCB |
|---|---|---|---|---|---|---|---|---|
| TabRepo [RS] | p-value | **0.00000** | **0.00000** | **0.00000** | **0.00000** | **0.00000** | **0.00006** | **0.00000** |
| | w/t/l | 186/4/10 | 179/6/15 | 135/5/60 | 185/4/11 | 172/5/23 | 126/3/71 | 135/5/60 |
| TabRepoRaw [SMAC] | p-value | **0.00072** | **0.00261** | 0.95063 | 0.42777 | 0.99194 | 0.99984 | 0.99194 |
| | w/t/l | 24/0/6 | 23/0/7 | 11/0/19 | 16/0/14 | 9/0/21 | 6/0/24 | 9/0/21 |
| YaHPOGym [SMAC] | p-value | **0.00880** | **0.00503** | 0.31038 | **0.00074** | 0.08372 | 0.50000 | **0.02412** |
| | w/t/l | 64/0/39 | 65/0/38 | 54/1/48 | 68/0/35 | 59/0/44 | 52/0/51 | 62/0/41 |

Table 1: P-values from a sign test assessing whether bandit methods outperform *combined search*. P-values below $\alpha = 0.05$ are underlined, while those below $\alpha = 0.05$ after multiple comparison correction (adjusting $\alpha$ by #comparisons) are boldfaced, indicating that the two-level approach is superior to *combined search*. Additionally, we report the number of wins, ties, and losses (w/t/l).

## 6 Conclusions, Discussions and Future Work

This paper addresses the CASH problem, proposing MaxUCB, an MKB method. Our data-driven analysis answers an open question on the applicability of extreme bandits to CASH. We provide a novel theoretical analysis and show state-of-the-art performance on several important benchmarks.

**Limitations.** Though our method can be applied beyond AutoML in principle, it is finely tuned for this setting with bounded and skewed distributions and for maximal value optimization. Our analysis relies on stationary distributions, which might not always be accurate, especially at the beginning of the HPO run, so a short burn-in phase may be needed to reach this regime. Our approach may not be distributionally optimal, as optimality in bounded extreme bandits remains an open question, and establishing lower bounds is left for future work. Lastly, we provide a default value for our hyperparameter $\alpha$ that might need adjusting for other applications.

**Impact on AutoML systems.** Our approach complements prior work on AutoML systems and increases their flexibility. First, our approach allows to choose any HPO method for each model at the lower level and thus it may integrate recent progress in HPO methods for some ML model classes, e.g., multi-fidelity or meta-learned methods [György and Kocsis, 2011, Li et al., 2018, Falkner et al., 2018, Müller et al., 2023, Chen et al., 2022]. Second, some AutoML systems [Swearingen et al., 2017, Li et al., 2023] decompose the search space into smaller subspaces to scale to a distributed setting and use Bandit methods to select promising subspaces [Levine et al., 2017, Li et al., 2020]. While we focus on applying bandits to select promising ML models, our methods could also be applied in this setting. Finally, beyond CASH, MaxUCB is well-suited for sub-supernet selection in Neural Architecture Search (NAS) [Hu et al., 2022], showing similar reward distributions [Ly-Manson et al., 2024] (see Appendix E.5).

**Choosing $\alpha$ adaptively.** Figure 5 suggests that one could try to tune $\alpha$ online. However, it is known that, in theory, without additional information, data-adaptive parameters cannot be found at a reasonable exploration cost in bandit optimization settings [Locatelli and Carpentier, 2018]. In AutoML systems, though, supplementary data, like estimates of the sub-optimality gap, reward distribution shape, and HPO convergence rate, can help adjust $\alpha$ adaptively.

**Future Directions.** To extend our extreme bandit setting, one could further refine the reward modeling by incorporating the non-stationarity, especially in AutoML for data streams, where optimal models shift with data distributions [Verma et al., 2024]. Incorporating cost-aware optimization is another promising direction, as computational resources and time, rather than iteration counts, often define budgets in AutoML; this would require estimating model training times and factoring them into the decision process. Addressing the growing complexity of heterogeneous ML tasks, such as those involving pre-training, fine-tuning, or prompt engineering, may benefit from a hierarchical approach that allocates resources effectively across diverse [Balef and Eggensperger, 2025a,b]. Additionally, exploiting structural similarities among algorithms and their hyperparameters could reduce exploration costs by sharing information across arms in the CASH problem, further enhancing efficiency.

## Acknowledgments and Disclosure of Funding

The authors are funded by the Deutsche Forschungsgemeinschaft (DFG, German Research Foundation) under Germany's Excellence Strategy – EXC number 2064/1 – Project number 390727645. Additionally, C. Vernade acknowledges funding from the DFG under the project 468806714 of the Emmy Noether Programme. The authors also thank the International Max Planck Research School for Intelligent Systems (IMPRS-IS).

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

# Table of Contents for the Appendices

## A   Preliminiaries

**Lemma A.1.** *Let $X_1, \ldots, X_n$ be $n$ samples independently drawn from distribution d, and let $G(x) = P(X > x)$ be the survival function. We have:*

$$P\left(\max_{1 \leq t \leq n} X_t \leq x\right) \leq e^{-nG(x)},$$

$$P\left(\max_{1 \leq t \leq n} X_t > x\right) \leq nG(x). \tag{10}$$

*Proof.* Let $F(x) = P(X \leq x)$ be the cumulative distribution function, so $G(x) = 1 - F(x) = P(X > x)$. First, consider the probability that the maximum of the $n$ samples is less than or equal to $x$:

$$P\left(\max_{1 \leq t \leq n} X_t \leq x\right) = \prod_{i=1}^{n} P(X_i \leq x) = (F(x))^n = (1 - G(x))^n \leq e^{-nG(x)}, \tag{11}$$

using the inequality $(1 - x)^n \leq e^{-nx}$. Next, consider the probability that the maximum of the $n$ samples is greater than $x$:

$$P\left(\max_{1 \leq t \leq n} X_t > x\right) \leq \sum_{i=1}^{n} P(X > x) = nG(x). \tag{12}$$

$\square$

## B   Proofs

### B.1   Proof of Lemma 3.3

**Lemma 3.3.** Suppose Assumption 3.2 holds. Then, there exists $L, U \geq 0$ such that the survival function $G$ can be bounded near $b$ by two linear functions.

$$\forall \epsilon \in (0, b - a), \quad L\epsilon \leq G(b - \epsilon) \leq U\epsilon \tag{13}$$

*Proof.* By applying the Mean Value Theorem (MVT) to the survival function $G$ over an interval $[b - \epsilon, b]$, there exists a point $c \in (b - \epsilon, b)$ such that:

$$G(b) - G(b - \epsilon) = G'(c)(b - (b - \epsilon)) = G'(c)\epsilon \tag{14}$$

Since $G'(x) = -f(x)$ where $f(x)$ the probability density function (PDF) and $G(b) = 0$ we have:

$$G(b - \epsilon) = f(c)\epsilon \tag{15}$$

Let $L$ and $U$ be the minimum and maximum values of the PDF $f(c) = \frac{G(b-\epsilon)}{\epsilon}$ over $\epsilon \in (0, b - a)$.

$$L\epsilon \leq G(b - \epsilon) \leq U\epsilon \tag{16}$$

Notably, in cases where we are interested in the survival function near some $b_1 < b$ and $a_1 > a$ i.e., over $(b_1 - \epsilon, b_1)$ applying the MVT again, there exists $c \in (b_1 - \epsilon, b_1)$ such that:

$$G(b_1) - G(b_1 - \epsilon) = G'(c)\epsilon = -f(c)\epsilon \tag{17}$$

which rearranges to:

$$G(b_1 - \epsilon) = f(c)\epsilon + G(b_1) \tag{18}$$

For any $\epsilon \in (\delta, b_1 - a_1)$ with $\delta > 0$ we can bound $G(b_1 - \epsilon)$ as:

$$L\epsilon \leq L\epsilon + G(b_1) \leq G(b_1 - \epsilon) \leq (f(c) + \frac{G(b_1)}{\delta})\epsilon \leq U\epsilon \tag{19}$$

$\square$

## B.2 Proof of Proposition 4.1

**Proposition 4.1.** (Upper Regret Bound) the upper regret bound up to time $T$ is related to the number of times sub-optimal arms are pulled,

$$R(T) \leq \frac{\max_{i \leq K} b_i}{T} \sum_{i \neq i^*}^{K} N_i(T) \tag{20}$$

Where $N_i(T) = \mathbb{E}(\sum_{t=1}^{T} \mathbb{1}\{I_t = i\})$ is the number of sub-optimal pulls of arm $i$, arm $i^*$ is the optimal arm and $b_i$ is the upper bound on the reward of arm $i$ as given by Assumption 3.2, i.e., the reward of arm $i$ lies within the interval $[a_i, b_i]$.

*Proof.* This proof is inspired by Assumption 1 of Baudry et al. [2022]. First, we need to determine an upper bound for the difference in the highest observed reward for arm $i$ when it has been pulled for $N_i(T)$ times compared to when it has been pulled for $T$ times.

$$\mathbb{E}\left[\max_{t \leq T} r_{i,t}\right] - \mathbb{E}\left[\max_{t \leq N_i(T)} r_{i,t}\right] = \mathbb{E}\left[\mathbb{1}\left\{\max_{N_i(T)+1 \leq t \leq T} r_{i,t} = \max_{t \leq T} r_{i,t}\right\} \max_{N_i(T)+1 \leq t \leq T} r_{i,t}\right]$$

$$\leq \mathbb{E}\left[\mathbb{1}\left\{\max_{N_i(T)+1 \leq t \leq T} r_{i,t} = \max_{t \leq T} r_{i,t}\right\} \underbrace{\mathbb{1}\left\{\max_{N_i(T)+1 \leq t \leq T} r_{i,t} \leq B\right\} \max_{N_i(T)+1 \leq t \leq T} r_{i,t}}_{\leq B}\right]$$

$$+ \mathbb{E}\left[\mathbb{1}\left\{\max_{N_i(T)+1 \leq t \leq T} r_{i,t} = \max_{t \leq T} r_{i,t}\right\} \mathbb{1}\left\{\max_{N_i(T)+1 \leq t \leq T} r_{i,t} > B\right\} \max_{N_i(T)+1 \leq t \leq T} r_{i,t}\right]$$

$$\leq P\left(\max_{N_i(T)+1 \leq t \leq T} r_{i,t} = \max_{t \leq T} r_{i,t}\right) B$$

$$+ \mathbb{E}\left[\mathbb{1}\left\{\max_{N_i(T)+1 \leq t \leq T} r_{i,t} = \max_{t \leq T} r_{i,t}\right\} \mathbb{1}\left\{\max_{N_i(T)+1 \leq t \leq T} r_{i,t} > B\right\} \max_{N_i(T)+1 \leq t \leq T} r_{i,t}\right]. \tag{21}$$

Since always $\max_{N_i(T)+1 \leq t \leq T} r_{i,t} \leq b_i$ by choosing $B = b_i$ we ensure that $\mathbb{1}\left\{\max_{N_i(T)+1 \leq t \leq T} r_{i,t} > B\right\} = 0$, leading to:

$$\mathbb{E}\left[\max_{t \leq T} r_{i,t}\right] - \mathbb{E}\left[\max_{t \leq N_i(T)} r_{i,t}\right] \leq P\left(\max_{N_i(T)+1 \leq t \leq T} r_{i,t} = \max_{t \leq T} r_{i,t}\right) B$$

$$\leq \left(1 - \frac{N_i(T)}{T}\right) B = \left(1 - \frac{N_i(T)}{T}\right) b_i. \tag{22}$$

Using this and according to the regret definition, we obtain the following:

$$R(T) = \mathbb{E}[\max_{t \leq T} r_{i^*,t}] - \mathbb{E}[\max_{t \leq T} r_{I_t,t}]$$

$$\leq \mathbb{E}\left[\max_{t \leq T} r_{i^*,t}\right] - \max_{i \leq K} \mathbb{E}\left[\max_{t \leq N_i(T)} r_{i,t}\right]$$

$$= \min_{i \leq K}(\mathbb{E}\left[\max_{t \leq T} r_{i^*,t}\right] - \mathbb{E}\left[\max_{t \leq N_i(T)} r_{i,t}\right])$$

$$= \min_{i \leq K}(\Delta_i + \underbrace{\mathbb{E}\left[\max_{t \leq T} r_{i,t}\right] - \mathbb{E}\left[\max_{t \leq N_i(T)} r_{i,t}\right]}_{Equation\ 22})$$

$$\leq \min_{i \leq K}(\Delta_i + (1 - \frac{N_i(T)}{T})b_i)$$

$$\leq \min_{i \leq K}(\Delta_i + (1 - \frac{N_i(T)}{T}) \max_{i \leq K} b_i) \tag{23}$$

Based on Definition 3.1, the suboptimality gap for the optimal arm $i^*$ is zero ($\Delta_{i^*} = 0$). Additionally, the total number of pulls for the optimal arm $N_{i^*}(T)$ can be calculated as the difference between the total number of pulls across all arms $T$ and the pulls for all suboptimal arms ($i \neq i^*$), i.e. $N_{i^*}(T) = T - \sum_{i \neq i^*}^{K} N_i(T)$. We upper bound the min in equation 23 by the specific value for the optimal arm $i^*$:

$$R(T) \leq \frac{\max_{i \leq K} b_i}{T} \sum_{i \neq i^*}^{K} N_i(T) \tag{24}$$

$\square$

## B.3    Proof of Theorem 4.2

**Theorem 4.2.** For any suboptimal arm $i \neq i^\star$, the number of suboptimal draws $N_i(T)$ performed by **Algorithm 1** up to time $T$ is bounded by

$$N_i(T) \leq \frac{T^{1-2L_{i^*}\alpha\sqrt{\Delta_i}}}{1 - 2L_{i^*}\alpha\sqrt{\Delta_i}} + 2\alpha\sqrt{U_iT}\log(T) \tag{25}$$

*Proof.* In order to find the upper bound for the number of sub-optimal pulls of arm $i$ for the algorithm 1, without loss of generality, we assume that arm 1 is the optimal arm, i.e. $i^* = 1$. Let $\Delta_i = \mathbb{E}[\max_{t \leq T} r_{1,t}] - \mathbb{E}[\max_{t \leq T} r_{i,t}]$ be the suboptimality gap. Our goal is to determine an upper bound on $N_i(T)$, the number of times the sub-optimal arm $i$ has been pulled up to time $T$. First, we identify the event that the algorithm pulls the sub-optimal arm $i$ at time $t$:

$$S = \{\max(r_{1,1}, ..., r_{1,n_1(t)}) + C_t(n_1(t)) \leq \max(r_{i,1}, ..., r_{i,n_i(t)}) + C_t(n_i(t))\}$$

$$= \{\max_{1 \leq n \leq n_1(t)} r_{1,n} + C_t(n_1(t)) \leq \max_{1 \leq n \leq n_i(t)} r_{i,n} + C_t(n_i(t))\} \tag{26}$$

where $S$ is the event of selecting a sub-optimal arm $i$ with a padding function $C_t(n)$. The exploration bonus $C_t(n)$ is a function that is designed to account for the exploration-exploitation trade-off and typically depends on $t$ and the number of times each arm has been pulled $n$. Let $n_1(t)$ and $n_i(t)$ represent the number of times the optimal arm 1 and an sub-optimal arm $i$ have been pulled, respectively, where $n_1(t) \leq t$ and $n_i(t) \leq t$. We want to express $S$ in a union of events that covers all possible scenarios leading to $S$. Thus, we split $S$ into two complementary conditions as follows:

$$S \subseteq \{\max_{1 \leq n \leq n_1(t)} r_{1,n} + C_t(n_1(t)) \leq x\} \cup \{\max_{1 \leq n \leq n_i(t)} r_{i,n} + C_t(n_i(t)) > x\} \tag{27}$$

Where $x$ is a threshold value, we take $x = \mathbb{E}[\max_{t \leq T} r_{i,t}]$,

$$
\begin{aligned}
S \subseteq & \left\{ \max_{1 \leq n \leq n_1(t)} r_{1,n} + C_t(n_1(t)) \leq \mathbb{E}[\max_{t \leq T} r_{i,t}] \right\} \\
& \cup \left\{ \max_{1 \leq n \leq n_i(t)} r_{i,n} + C_t(n_i(t)) > \mathbb{E}[\max_{t \leq T} r_{i,t}] \right\} \\
= & \left\{ \max_{1 \leq n \leq n_1(t)} r_{1,n} \leq \mathbb{E}[\max_{t \leq T} r_{1,t}] - \Delta_i - C_t(n_1(t)) \right\} \\
& \cup \left\{ \max_{1 \leq n \leq n_i(t)} r_{i,n} > \mathbb{E}[\max_{t \leq T} r_{i,t}] - C_t(n_i(t)) \right\}
\end{aligned}
\tag{28}
$$

Thus, the event $S$ can be contained within the union of two bad events:

- Underestimating the upper confidence bound of extreme values for the optimal arm 1

- Overestimating the upper confidence bound of extreme values for the sub-optimal arm $i$

Now we use Lemma A.1 to calculate the probability of 28:

$$
\begin{aligned}
P(S) \leq\ & P\left( \left\{ \max_{1 \leq n \leq n_1(t)} r_{1,n} \leq \mathbb{E}[\max_{t \leq T} r_{1,t}] - C_t(n_1(t)) - \Delta_i \right\} \right) \\
& + P\left( \left\{ \max_{1 \leq n \leq n_i(t)} r_{i,n} > \mathbb{E}[\max_{t \leq T} r_{i,t}] - C_t(n_i(t)) \right\} \right) \\
\leq\ & e^{-n_1(t) G_1 \left( \mathbb{E}[\max_{t \leq T} r_{1,t}] - C_t(n_1(t)) - \Delta_i \right)} \\
& + n_i(t) G_i \left( \mathbb{E}[\max_{t \leq T} r_{i,t}] - C_t(n_i(t)) \right).
\end{aligned}
\tag{29}
$$

Now, by applying Lemma 3.3, we can simplify the analysis by eliminating the complexities associated with survival functions $G_1$ and $G_i$.

$$
P(S) \leq e^{-n_1(t) L_1 (C_t(n_1(t)) + \Delta_i)} + n_i(t) U_i C_t(n_i(t))
\tag{30}
$$

For the first term of the right-hand side, by the Arithmetic Mean-Geometric Mean (AM-GM) inequality ($a + b \geq 2\sqrt{ab}$) of equation 30, we have:

$$
e^{-n_1(t) L_1 (C_t(n_1(t)) + \Delta_i)} \leq e^{-2L_1 n_1(t) \sqrt{C_t(n_1(t)) \Delta_i}}
\tag{31}
$$

In this stage, we want to find a proper padding function $C_t(n)$, which controls the right-hand side of equation 30 and equation 31. By choosing $C_t(n) = (\frac{\alpha \log(t)}{n})^2$, we have:

$$
e^{-n_1(t) L_1 (C_t(n_1(t)) + \Delta_i)} \leq e^{-2L_1 n_1(t) \sqrt{C_t(n_1(t)) \Delta_i}} = e^{-2L_1 \alpha \log(t) \sqrt{\Delta_i}} = t^{-2L_1 \alpha \sqrt{\Delta_i}}
\tag{32}
$$

$$
n_i(t) U_i C_t(n_i(t)) \leq \frac{\alpha^2 U_i \log^2(t)}{n_i(t)}
\tag{33}
$$

This selection of the function of the exploration bonus results in two significant advantages. First, it provides an upper bound for the right-hand side of equation 30 that remains independent of $n_1$. Furthermore, Equation 31 shows a decreasing trend as the number of pulls for the sub-optimal arm $i$ increases.[8]

---

[8]We note that this choice is not based on the inherent property of maximum values. In general one can use $C_t(n) = (\frac{\alpha \log(t)}{n})^m$ for $m > 1$ with the optimal $m$ depending on the setting. In Appendix B.4 we show how $m$ affects the regret.

Now, we assume that the sub-optimal arm $i$ has been played for $l_i$ times, so $n_i(t) \geq l_i$. We want to calculate the number of sub-optimal pulls of arm $i$ up to time $T$:

$$N_i(T) \leq l_i + \sum_{t=l_i}^{T} P(S) \leq l_i + \sum_{t=l_i}^{T} t^{-2L_1\alpha\sqrt{\Delta_i}} + \sum_{t=l_i}^{T} \frac{\alpha^2 U_i \log^2(t)}{l_i}$$

$$\leq l_i + \frac{T^{1-2L_1\alpha\sqrt{\Delta_i}}}{1 - 2L_1\alpha\sqrt{\Delta_i}} + \frac{\alpha^2 U_i}{l_i} T \log^2(T) \tag{34}$$

By choosing $l_i = \alpha\sqrt{U_i T}\log(T)$, we have:

$$N_i(T) \leq \frac{T^{1-2L_1\alpha\sqrt{\Delta_i}}}{1 - 2L_1\alpha\sqrt{\Delta_i}} + 2\alpha\sqrt{U_i T}\log(T) \tag{35}$$

$\square$

## B.4  Extension of Proof of Theorem 4.2

As we discuss in Section 3, the reward distribution in our setting is left-skewed. We now show that under the assumption that the survival function decays rapidly near the maximum i.e., $G_i(\mathbb{E}[\max_{t\leq T} r_{i,t}]) = \mathcal{O}(\frac{1}{T^2})$, a tighter bound can be derived. This assumption means that the reward distribution has a very light-tail near its upper extreme, which *may* often hold for our left-skewed distributions. Furthermore, we generalize our algorithm by assuming $C_t = (\frac{\alpha \log(t)}{n_i})^m$ as a exploration bonus function, where $m \geq 1$ is a hyperparameter.

*Proof.* We begin with Equation 27 and we take $x = \mathbb{E}[\max_{t\leq T} r_{i,t}] + c\Delta_i$, where $c$ is an arbitrary variable $c \in [0,1]$. We have:

$$S \subseteq \left\{ \max_{1\leq n\leq n_1(t)} r_{1,n} + C_t(n_1(t)) \leq \mathbb{E}[\max_{t\leq T} r_{i,t}] + c\Delta_i \right\}$$

$$\cup \left\{ \max_{1\leq n\leq n_i(t)} r_{i,n} + C_t(n_i(t)) > \mathbb{E}[\max_{t\leq T} r_{i,t}] + c\Delta_i \right\}$$

$$= \left\{ \max_{1\leq n\leq n_1(t)} r_{1,n} \leq \mathbb{E}[\max_{t\leq T} r_{1,t}] - (1-c)\Delta_i - C_t(n_1(t)) \right\}$$

$$\cup \left\{ \max_{1\leq n\leq n_i(t)} r_{i,n} > \mathbb{E}[\max_{t\leq T} r_{i,t}] - C_t(n_i(t)) + c\Delta_i \right\}$$

$$= \underbrace{\left\{ \max_{1\leq n\leq n_1(t)} r_{1,n} \leq \mathbb{E}[\max_{t\leq T} r_{1,t}] - (1-c)\Delta_i - C_t(n_1(t)) \right\}}_{S_1} \tag{36}$$

$$\cup \underbrace{\left\{ \max_{1\leq n\leq n_i(t)} r_{i,n} > \mathbb{E}[\max_{t\leq T} r_{i,t}] - C_t(n_i(t)) + c\Delta_i, \quad C_t(n_i(t)) \leq c\Delta_i \right\}}_{S_2} \tag{37}$$

$$\cup \underbrace{\left\{ \max_{1\leq n\leq n_i(t)} r_{i,n} > \mathbb{E}[\max_{t\leq T} r_{i,t}] - C_t(n_i(t)) + c\Delta_i, \quad C_t(n_i(t)) > c\Delta_i \right\}}_{S_3} \tag{38}$$

First, we calculate the probability of event $S_2$:

$$P(S_2) \leq P(\left\{ \max_{1\leq n\leq n_i(t)} r_{i,n} > \mathbb{E}[\max_{t\leq T} r_{i,t}] \right\}) \leq n_i G_i(\mathbb{E}[\max_{t\leq T} r_{i,t}]) \leq T G_i(\mathbb{E}[\max_{t\leq T} r_{i,t}]). \tag{39}$$

By calculating the number of sub-optimal pulls of arm $i$ up to time $T$, we know the third part of the event ($S_3$) can happen at most $C_T^{-1}(c\Delta_i) = (\frac{\alpha \log(T)}{c\Delta_i})^{\frac{1}{m}}$ times:

$$N_i(T) \leq (\frac{\alpha \log(T)}{c\Delta_i})^{\frac{1}{m}} + \sum_{t=1}^{T} P(S_1) + \sum_{t=1}^{T} P(S_2) \tag{40}$$

$$\leq (\frac{\alpha \log(T)}{c\Delta_i})^{\frac{1}{m}} + T^2 G_i(\mathbb{E}[\max_{t \leq T} r_{i,t}]) + \sum_{t=1}^{T} P(S_1) \tag{41}$$

$$\leq (\frac{\alpha \log(T)}{c\Delta_i})^{\frac{1}{m}} + M + \sum_{t=1}^{T} P(S_1) \tag{42}$$

Where $M$ is a constant as we assume $G_i(\mathbb{E}[\max_{t \leq T} r_{i,t}]) = \mathcal{O}(\frac{1}{T^2})$. Finally, we need to find an upper bound for $P(S_1)$. We need to differentiate between two situations, when $m = 1$ and when $m > 1$

**For $m = 1$.** We set $c = 1$ and then we have:

$$P(S_1) \leq e^{-n_1 G_1(\mathbb{E}[\max_{t \leq T} r_{1,t}] - (1-c)\Delta_i - C_t(n_1(t)))} \leq \tag{43}$$

$$e^{-L_1(C_t(n_1(t)) + (1-c)\Delta_i)} \leq e^{-\alpha L_1 \log(T)} \leq T^{-\alpha L_1} \tag{44}$$

Finally, we have:

$$N_i(T) \leq M + \frac{\alpha \log(T)}{\Delta_i} + \frac{T^{1-\alpha L_1}}{1 - \alpha L_1} \tag{45}$$

With $\alpha > \frac{1}{L_1}$:

$$N_i(T) = \mathcal{O}(\frac{\log(T)}{L_1 \Delta_i}) \tag{46}$$

**For $m > 1$.** We know $n((\frac{a}{n})^m + b) \geq ab^{\frac{m-1}{m}}[m(m-1)^{\frac{1}{m}-1}]$. We have:

$$P(S_1) \leq e^{-n_1(t) L_1(C_t(n_1(t)) + (1-c)\Delta_i)} \leq e^{-\alpha L_1 \log(T)((1-c)\Delta_i)^{\frac{m-1}{m}}[m(m-1)^{\frac{1}{m}-1}]} \tag{47}$$

$$\leq T^{-\alpha L_1((1-c)\Delta_i)^{\frac{m-1}{m}}[m(m-1)^{\frac{1}{m}-1}]} \tag{48}$$

For simplicity, we set $c = \frac{1}{2}$. Then:

$$N_i(T) \leq (\frac{\alpha \log(T)}{c\Delta_i})^{\frac{1}{m}} + M + \sum_{t=1}^{T} P(S_2) \tag{49}$$

$$\leq (\frac{2\alpha \log(T)}{\Delta_i})^{\frac{1}{m}} + M + \frac{T^{1-\alpha L_1(\frac{\Delta_i}{2})^{\frac{m-1}{m}}[m(m-1)^{\frac{1}{m}-1}]}}{1 - \alpha L_1(\frac{\Delta_i}{2})^{\frac{m-1}{m}}[m(m-1)^{\frac{1}{m}-1}]} \tag{50}$$

Finally, we have:

$$N_i(T) = (\frac{2\alpha \log(T)}{\Delta_i})^{\frac{1}{m}} + \mathcal{O}(T^{1-\alpha L_1(\frac{\Delta_i}{2})^{\frac{m-1}{m}}[m(m-1)^{\frac{1}{m}-1}]}). \tag{51}$$

And by choosing $\alpha$ as below

$$\alpha = \mathcal{O}\left(\frac{1}{L_1(\Delta_i)^{\frac{m-1}{m}}}\right), \tag{52}$$

We have:

$$N_i(T) = \mathcal{O}\left(\frac{\log(T)}{L_1 \Delta_i^{\frac{2m-1}{m}}}\right)^{\frac{1}{m}}.$$

(53)

$\square$

We would like to emphasize again that this result only holds when $G_i(\mathbb{E}[\max_{t \leq T} r_{i,t}])$ is sufficiently small. By using a weaker assumption, controlling $G_i(\mathbb{E}[\max_{t \leq T} r_{i,t}])$ is necessary. In Theorem B.3, we address this for our method by considering $C_t(n_i(t)) \geq G_i(\mathbb{E}[\max_{t \leq T} r_{i,t}])$ to ensure proper control.

Notably, in general, $\Delta_i$ depends on $T$ (see Definition 3.1). Meaning that $\frac{\log(T)}{\Delta_i}$ does not necessarily lead to a logarithmic regret. However, in some special scenarios where the reward distributions have different supports, we can ensure that $\Delta_i = b_1 - b_i$ as $T$ approaches infinity, and a logarithmic regret is achievable.

Furthermore, Equation 53 shows that the parameter $m$ controls $N_i(T)$, the number of times a suboptimal arm is pulled asymptotically. When $T$ is very large, a higher value of $m$ (along with the optimal $\alpha$) improves performance. However, it makes the algorithm more sensitive to the choice of $\alpha$. As shown in Equation 52, with increasing $m$, $\Delta_i$ has a greater influence on the optimal $\alpha$. Noting that $\Delta_i$ varies across arms and is typically unknown for an unseen task, finding the optimal $\alpha$ is not feasible in practice. Therefore, there is a trade-off between performance and sensitivity. In the CASH setting, $m = 2$ performs empirically well, while exhibiting low sensitivity to $\alpha$. Furthermore, as shown in Appendix D.4, we can find a range of $\alpha$ for which the MaxUCB works well across different CASH tasks.

# C  More Details on Reward Distribution

## C.1  Reward distribution analysis

In addition to the analysis in the main paper in Figure 2 in Section 3, we collected the observed rewards (the output of HPO) for all arms on each model class. We calculate each dataset's empirical survival function $G$ and provide the reward distribution analysis for all benchmark tasks. Notably, the shift in distribution (indicated by the thin lines) is low for all tasks, not contradicting the i.i.d. assumptions. For our method, we design our algorithm based on analyzing the distribution of raw rewards (in contrast to the distribution of maximum values over time).

Over time, the maximum value of samples generated from an i.i.d. distribution has an increasing trend, i.e., the extreme values get better. Notably, this is not contradictory with the *Rising Bandits* strategy Liu et al. [2019], which assumes that the maximum observed value over time is not decreasing (and then analyses the trend of this maximum observed value as a (non i.i.d) reward).

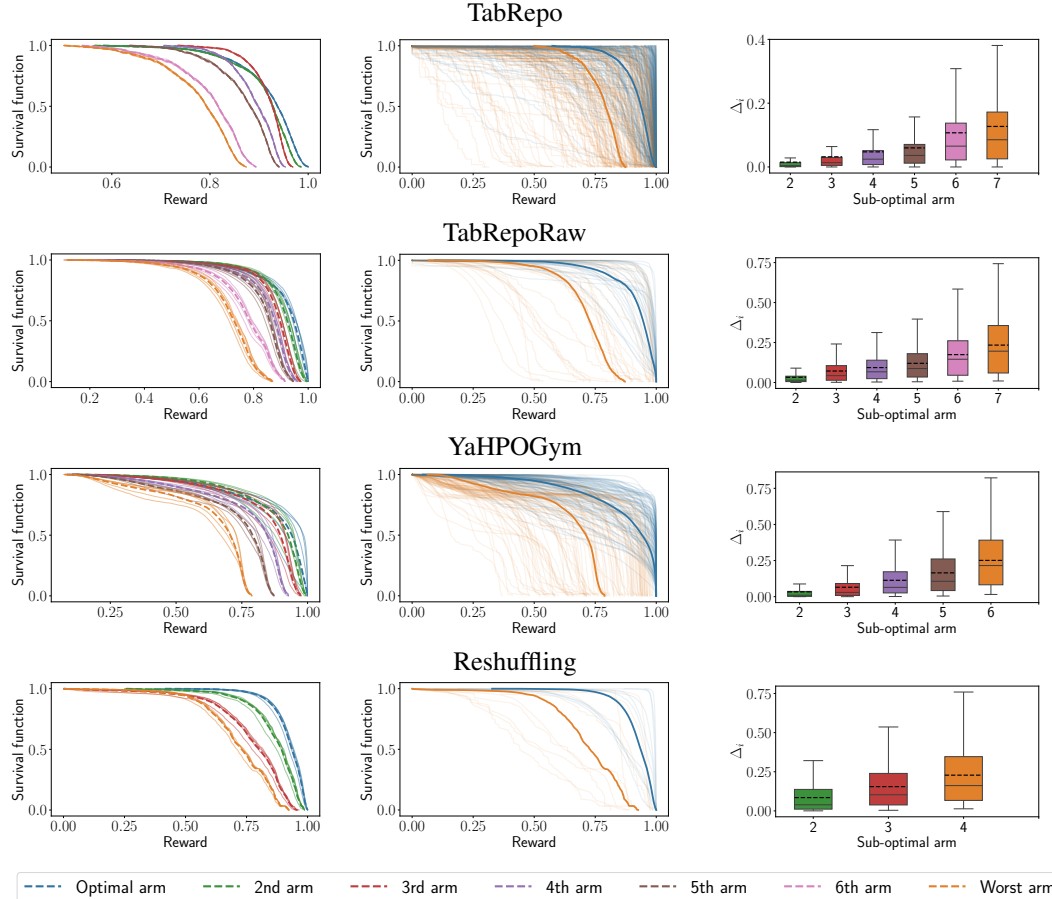

Figure C.1: (Left) The average empirical survival function of rewards (observed performances) per arm ranked per dataset. We divided the reward sequence into five segments over the budget (time horizon) to show the distribution change over time. Thin lines correspond to empirical survival functions for different segments, visualizing the change over time. (Middle) The average empirical survival function per dataset for the best and worst arm with thin lines corresponding to individual datasets. (Right) The sub-optimality gap $\Delta_i$.

## C.2 More Details on Lemma 3.3

$L$ and $U$ are lower and upper bounds for the tangent line approximation of the survival function $G$ near the maximum value, indicating the shape of the distribution. We provide three examples to demonstrate this numerically.

**Toy example**: Assume two simple survival functions $G_1(x) = 1 - x^2$ (left skewed, blue curve) and $G_2(x) = (1 - x)^2$ (right skewed, orange curve) with support $[0, 1]$. We calculate $G(1 - \epsilon)/\epsilon$ over some values of $\epsilon$ in the Figure C.2. To compute L and U, we determine the minimum and maximum values of $G(1 - \epsilon)/\epsilon$ over the range $0 < \epsilon < 1$. For clarity, we restricted $\epsilon$ to iterate only over the set $\{0.1, 0.3, 0.5, 0.7, 0.9\}$. It implies that the calculated values of $L$ and $U$ are valid for $\epsilon \in [0.1, 0.9]$.

| $\epsilon$ | $\frac{G_1(1-\epsilon)}{\epsilon}$ | $\frac{G_2(1-\epsilon)}{\epsilon}$ |
|---|---|---|
| 0.10 | 1.90 | 0.10 |
| 0.30 | 1.70 | 0.30 |
| 0.50 | 1.50 | 0.50 |
| 0.70 | 1.30 | 0.70 |
| 0.90 | 1.10 | 0.90 |
| L | 1.10 | 0.10 |
| U | 1.90 | 0.90 |

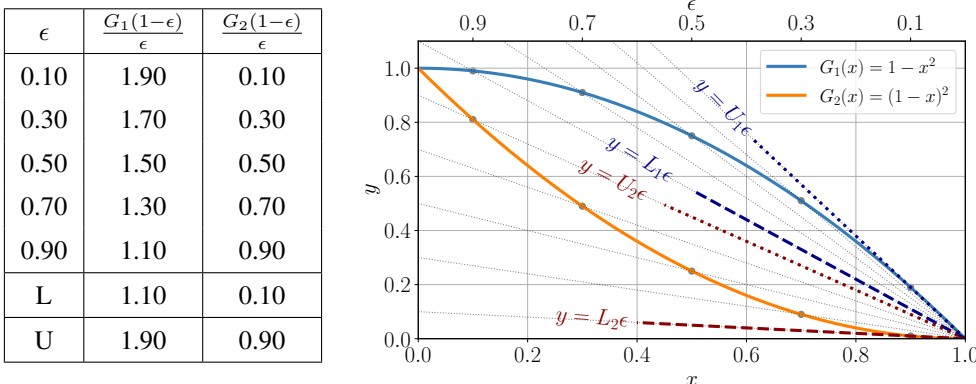

Figure C.2: (Left) Determining $L$ and $U$ for $G_1$ and $G_2$. We calculate $\frac{G(1-\epsilon)}{\epsilon}$ over different ranges of $\epsilon$. The minimum and maximum values obtained from this ratio are assigned as $L$ and $U$, respectively. (Right) Showing two survival functions $G_1$ and $G_2$ along with their linear line approximations (gray lines). These tangent lines illustrate how $L$ and $U$ effectively bound the survival function $G$ near its maximum value.

**Truncated uniform distribution**: Assume that we have a truncated uniform distribution with support $[a, b]$. We know $G(x) = \frac{b-x}{b-a}$ for $x \in (a, b)$. For every $\epsilon \in [a, b]$ we have $\frac{G(b-\epsilon)}{\epsilon} = \frac{1}{b-a}$, which means $L = U = \frac{1}{b-a}$.

**Truncated Gaussian distribution**: There is no closed-form solution to formulate $L$ and $U$ based on the parameters of the truncated Gaussian distribution. Thus, we show the results of simulations to estimate $L$ and $U$ for truncated Gaussian within $[0, 1]$ with various values $\mu$ and $\sigma$, averaging over 1000 runs in Figure C.3.

| $\mu$ | $\sigma$ | $L$ | $U$ |
|---|---|---|---|
| 0.25 | 0.5 | $0.58 \pm 0.06$ | $1.70 \pm 0.48$ |
| 0.50 | 0.5 | $0.85 \pm 0.07$ | $1.53 \pm 0.34$ |
| 0.75 | 0.5 | $1.01 \pm 0.01$ | $1.64 \pm 0.25$ |
| 0.25 | 0.2 | $0.34 \pm 0.07$ | $1.54 \pm 0.28$ |
| 0.50 | 0.2 | $0.44 \pm 0.08$ | $1.36 \pm 0.04$ |
| 0.75 | 0.2 | $1.01 \pm 0.00$ | $1.95 \pm 0.04$ |

Figure C.3: (Left) Determining L and U for truncated Gaussian within $[0, 1]$ with different values for $\mu$ and $\sigma$. Averaged over 1 000 runs. (Right) Showing survival function of truncated Gaussian distribution with different values for $\mu$ and $\sigma$.

**TabRepo**

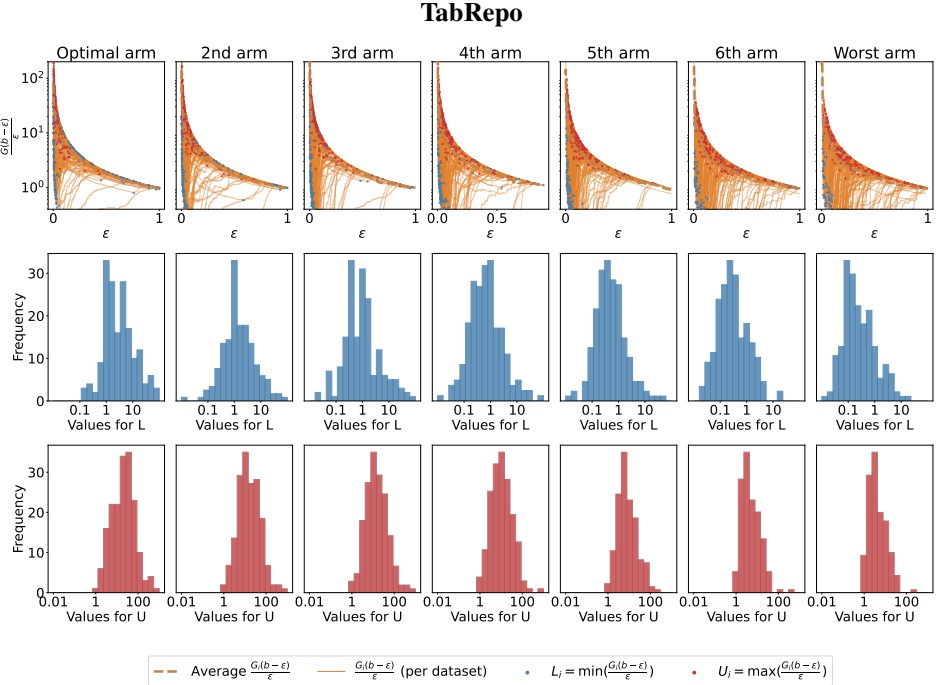

Figure C.4: Arms are ordered by sub-optimality gap. (Top) Thin orange lines represent $\frac{G(b-\epsilon)}{\epsilon}$, while the blue and red points correspond to $L$ and $U$ for our empirical reward distributions (see Lemma 3.3 for details). (Middle) Histogram of values for $L$. (Bottom) Histogram of values for $U$.

## C.3 Empirical Validation of Lemma 3.3

To study the empirical values for $L$ and $U$ in Lemma 3.3, we leverage the calculated empirical survival function $G$ for each dataset in Appendix C.1. Specifically, we evaluate $\frac{G(b-\epsilon)}{\epsilon}$ over the range $G^{-1}(0.99) < \epsilon < G^{-1}(0.01)$, where $G^{-1}(x)$ denotes the inverse of the survival function $G(x)$. Focusing on this range allows us to achieve a more robust estimation. Additionally, for TabRepo dataset, we exclude 23 datasets containing an arm with a standard deviation smaller than 0.001, further enhancing the robustness of our analysis. In Figures C.4, C.5, C.6, and C.7, the evaluated values of $\frac{G(b-\epsilon)}{\epsilon}$ for different benchmarks are shown. Additionally, the values for $L$ and $U$, corresponding to $\min_{\epsilon}(\frac{G(b-\epsilon)}{\epsilon})$ and $\max_{\epsilon}(\frac{G(b-\epsilon)}{\epsilon})$, respectively, are presented. Finally, the histograms of these two variables are also included.

**TabRepoRaw**

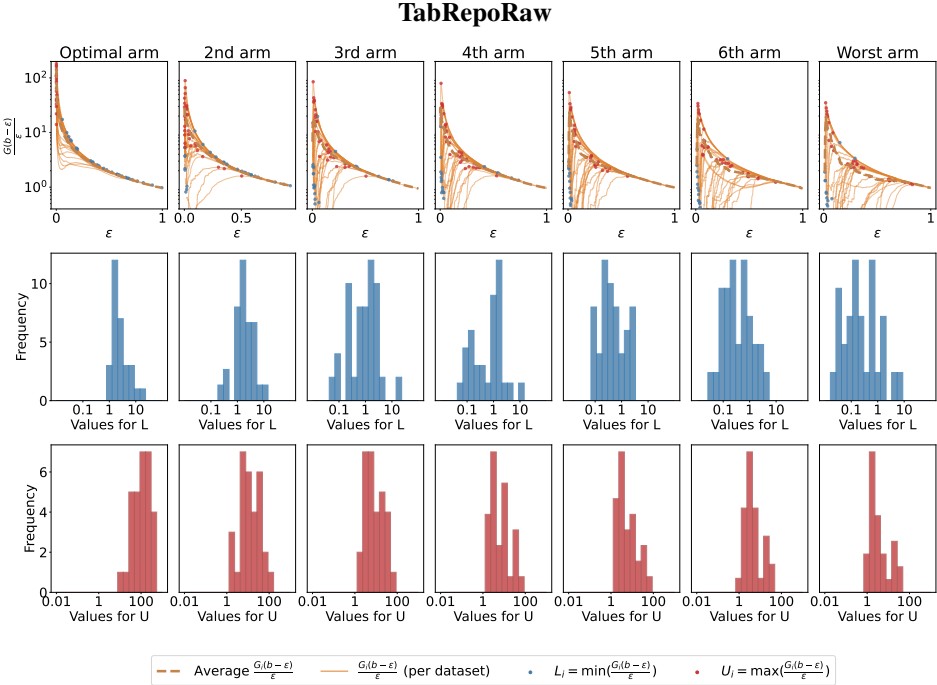

Figure C.5: Arms are ordered by sub-optimality gap. (Top) Thin orange lines represent $\frac{G(b-\epsilon)}{\epsilon}$, while the blue and red points correspond to $L$ and $U$ for our empirical reward distributions (see Lemma 3.3 for details). (Middle) Histogram of values for $L$. (Bottom) Histogram of values for $U$.

**YaHPOGym**

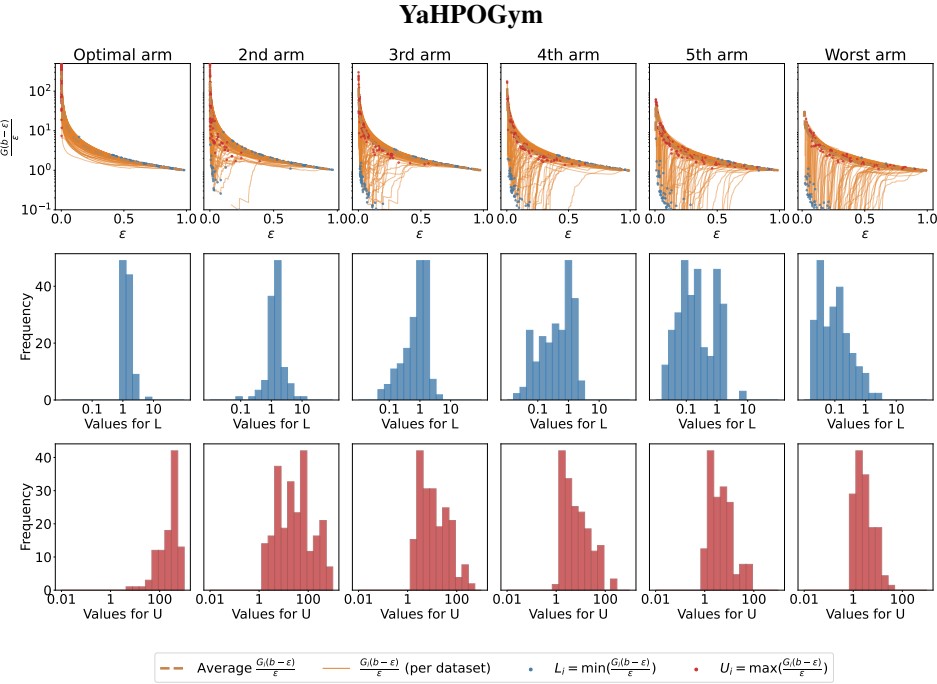

Figure C.6: Arms are ordered by sub-optimality gap. (Top) Thin orange lines represent $\frac{G(b-\epsilon)}{\epsilon}$, while the blue and red points correspond to $L$ and $U$ for our empirical reward distributions (see Lemma 3.3 for details). (Middle) Histogram of values for $L$. (Bottom) Histogram of values for $U$.

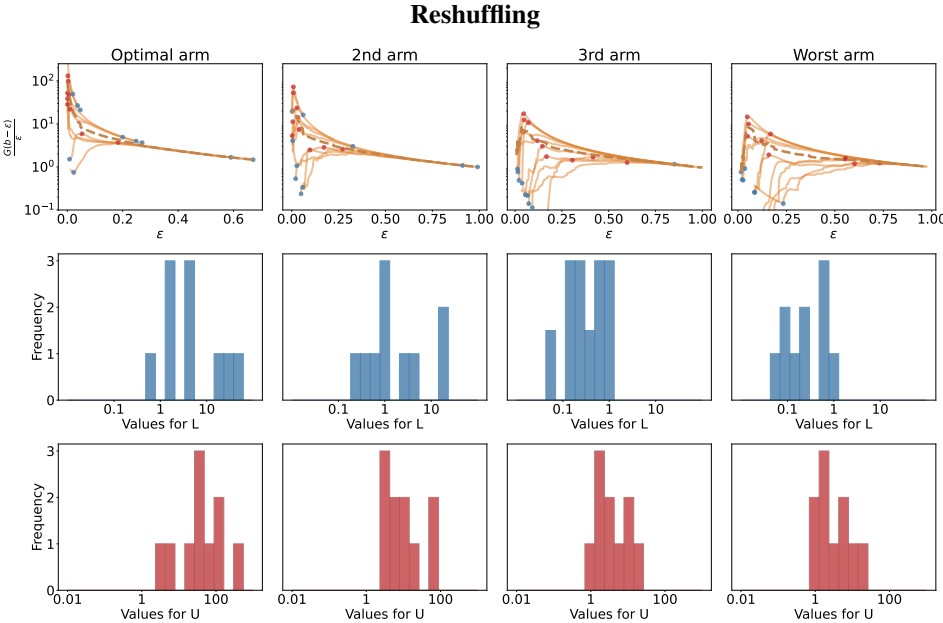

Figure C.7: Arms are ordered by sub-optimality gap. (Top) Thin orange lines represent $\frac{G(b-\epsilon)}{\epsilon}$, while the blue and red points correspond to $L$ and $U$ for our empirical reward distributions (see Lemma 3.3 for details). (Middle) Histogram of values for $L$. (Bottom) Histogram of values for $U$.

| name | #models | #tasks | type | HPO meth. (rep.) | budget | reference |
|------|---------|--------|------|------------------|--------|-----------|
| YaHPOGym | 6 | 103 | surrogate | *SMAC* (32) | 200 | [Pfisterer et al., 2022] |
| TabRepo | 7 | 200 | tabular | *random search* (32) | 200 | [Salinas and Erickson, 2024] |
| TabRepoRaw | 7 | 30 | raw | *SMAC* (32) | 200 | - |
| Reshuffling | 4 | 10 | raw | HEBO (30) | 250 | [Nagler et al., 2024] |

Table D.1: Overview of AutoML tasks. For TabRepo and Reshuffling, we use pre-computed HPO trajectories. TabRepoRaw resembles the same model space as TabRepo, but instead of *random search*, we run HPO ourselves. Similarly, we run HPO across provided surrogate HPO benchmark tasks YaHPOGym. We use SMAC [Lindauer et al., 2022, Hutter et al., 2011], implementing Bayesian optimization using Random forests for both tasks.

# D    More Details on the Experiments

## D.1    Metric calculation

**Average ranking calculation.** We use bootstrapping with Monte Carlo sampling to calculate the average ranking plot with confidence intervals. For each time step and each task in every dataset, we resample the performance of each repetition (with replacement) and compute the average performance. We then rank the algorithms based on these averaged performances and repeat this process for all tasks. Finally, we average the rankings across tasks. This entire procedure was repeated 1000 times to estimate the confidence interval.

**Number of wins, ties, and losses.** To determine the number of wins, ties, and losses for each task in every dataset, we first compute the average performance of each algorithm over all repetitions at the final time step. We then perform pairwise comparisons of these averaged performances among all algorithms versus *combined search*. To account for negligible differences that are not statistically significant, we consider two performances to be tied if they are sufficiently close. Specifically, we use NumPy's `isclose` function to compare the averaged performances, treating values within a default tolerance of $1 \times 10^{-8}$ as equal.

## D.2    Experimental Setup

Here, we provide details on our experimental setups. We used several well-established and widely used benchmark sets (as described in Table D.1) that were developed to compare HPO methods. Each benchmark contained different datasets, tasks, and search spaces to ensure that our empirical distribution analysis was not limited to a single source or problem type.

**YaHPOGym** [Pfisterer et al., 2022], a surrogate benchmark, covers 6 ML models (details in Table D.2) on 103 datasets and uses a regression model (surrogate model) to predict performances for queried hyperparameter settings. We use Bayesian optimization as implemented by SMAC [Lindauer et al., 2022] using Random Forests to conduct HPO. Additionally, we compare our two-level approach to *combined search* using SMAC, SMAC without initial design (SMAC-no-init), and Random Search. We run 32 repetitions and use a budget of 200 iterations for each evaluation.

**TabRepo** [Salinas and Erickson, 2024] consists of pre-evaluated performance scores for 200 iterations of random search for 7 ML models (details in Table D.3) on 200 datasets (context name: $D244\_F3\_C1530\_200$). We run 32 repetitions and use a budget of 200 iterations for each task.

**TabRepoRaw** which uses the search space from TabRepo (details in Table D.3) and allows HPO to evaluate all configurations. For constructing TabRepo [Salinas and Erickson, 2024], each configuration was evaluated with a one-hour time limit and 8-fold cross-validation. To reduce computational requirements for TabRepoRaw, we reduced this to 5 minutes and 4-fold cross-validation, and we provide it for 30 datasets (context name: $D244\_F3\_C1530\_30$). We use Bayesian optimization as implemented by SMAC [Lindauer et al., 2022] using Random Forests to conduct HPO. Additionally, we compare our two-level approach to *combined search* using SMAC, Random Search. We run 32 repetitions and use a budget of 200 iterations for the mentioned task.

To enable a fair comparison, we always evaluate the default configuration for each model first and then allow SMAC to run an initial design of $50 - \#arms$ configurations in the upper level and $\frac{50}{\#arms-1}$ in the lower level.

Table D.2: Hyperparameter spaces for ML models in YaHPOGym.

| ML model | Hyperparameter | Type | Range | Info |
|---|---|---|---|---|
| - | trainsize | continuous | [0.03, 1] | =0.525 (fixed) |
| | imputation | categorical | {mean, median, hist} | =mean (fixed) |
| Glmnet | alpha | continuous | [0, 1] | |
| | s | continuous | [0.001, 1097] | log |
| Rpart | cp | continuous | [0.001, 1] | log |
| | maxdepth | integer | [1, 30] | |
| | minbucket | integer | [1, 100] | |
| | minsplit | integer | [1, 100] | |
| SVM | kernel | categorical | {linear, polynomial, radial} | |
| | cost | continuous | [4.5e-05, 2.2e4] | log |
| | gamma | continuous | [4.5e-05, 2.2e4] | log, kernel |
| | tolerance | continuous | [4.5e-05, 2] | log |
| | degree | integer | [2, 5] | kernel |
| AKNN | k | integer | [1, 50] | |
| | distance | categorical | {l2, cosine, ip} | |
| | M | integer | [18, 50] | |
| | ef | integer | [7, 403] | log |
| | ef_construction | integer | [7, 403] | log |
| Ranger | num.trees | integer | [1, 2000] | |
| | sample.fraction | continuous | [0.1, 1] | |
| | mtry.power | integer | [0, 1] | |
| | respect.unordered.factors | categorical | {ignore, order, partition} | |
| | min.node.size | integer | [1, 100] | |
| | splitrule | categorical | {gini, extratrees} | |
| | num.random.splits | integer | [1, 100] | splitrule |
| XGBoost | booster | categorical | {gblinear, gbtree, dart} | |
| | nrounds | integer | [7, 2980] | log |
| | eta | continuous | [0.001, 1] | log, booster |
| | gamma | continuous | [4.5e-05, 7.4] | log, booster |
| | lambda | continuous | [0.001, 1097] | log |
| | alpha | continuous | [0.001, 1097] | log |
| | subsample | continuous | [0.1, 1] | |
| | max_depth | integer | [1, 15] | booster |
| | min_child_weight | continuous | [2.72, 148.4] | log, booster |
| | colsample_bytree | continuous | [0.01, 1] | booster |
| | colsample_bylevel | continuous | [0.01, 1] | booster |
| | rate_drop | continuous | [0, 1] | booster |
| | skip_drop | continuous | [0, 1] | booster |

**Reshuffling** [Nagler et al., 2024] which uses Heteroscedastic and Evolutionary Bayesian Optimization solver (HEBO) [Cowen-Rivers et al., 2022] for HPO. This benchmark includes HPO runs for 4 ML models (details in Table D.4) across 10 datasets, with 10 repetitions and 3 different validation split ratios within a budget of 250 iterations. Although the benchmark does not support HPO over the entire search space, it offers a valuable opportunity to compare the performance of bandit methods in a realistic setting.

Table D.3: Hyperparameter spaces for ML models in TabRepo and TabRepoRaw.

| ML model | Hyperparameter | Type | Range | Info | Default value |
|---|---|---|---|---|---|
| NN(PyTorch) | learning rate | continuous | [1e-4, 3e-2] | log | 3e-4 |
| | weight decay | continuous | [1e-12, 0.1] | log | 1e-6 |
| | dropout prob | continuous | [0, 0.4] | | 0.1 |
| | use batchnorm | categorical | False, True | | |
| | num layers | integer | [1, 5] | | 2 |
| | hidden size | integer | [8, 256] | | 128 |
| | activation | categorical | relu, elu | | |
| NN(FastAI) | learning rate | continuous | [5e-4, 1e-1] | log | 1e-2 |
| | layers | categorical | [200], [400], [200, 100], [400, 200], [800, 400], [200, 100, 50], [400, 200, 100] | | |
| | emb drop | continuous | [0.0, 0.7] | | 0.1 |
| | ps | continuous | [0.0, 0.7] | | 0.1 |
| | bs | categorical | 256, 128, 512, 1024, 2048 | | |
| | epochs | integer | [20, 50] | | 30 |
| CatBoost | learning rate | continuous | [5e-3 ,0.1] | log | 0.05 |
| | depth | integer | [4, 8] | | 6 |
| | l2 leaf reg | continuous | [1, 5] | | 3 |
| | max ctr complexity | integer | [1, 5] | | 4 |
| | one hot max size | categorical | 2, 3, 5, 10 | | |
| | grow policy | categorical | SymmetricTree, Depthwise | | |
| LightGBM | learning rate | continuous | [5e-3 ,0.1] | log | 0.05 |
| | feature fraction | continuous | [0.4, 1.0] | | 1.0 |
| | min data in leaf | integer | [2, 60] | | 20 |
| | num leaves | integer | [16, 255] | | 31 |
| | extra trees | categorical | False, True | | |
| XGBoost | learning rate | continuous | [5e-3 ,0.1] | log | 0.1 |
| | max depth | integer | [4, 10] | | 6 |
| | min child weight | continuous | [0.5, 1.5] | | 1.0 |
| | colsample bytree | continuous | [0.5, 1.0] | | 1.0 |
| | enable categorical | categorical | False, True | | |
| Extra-trees | max leaf nodes | integer | [5000, 50000] | | |
| | min samples leaf | categorical | 1, 2, 3, 4, 5, 10, 20, 40, 80 | | |
| | max features | categorical | sqrt, log2, 0.5, 0.75, 1.0 | | |
| Random-forest | max leaf nodes | integer | [5000, 50000] | | |
| | min samples leaf | categorical | 1, 2, 3, 4, 5, 10, 20, 40, 80 | | |
| | max features | categorical | sqrt, log2, 0.5, 0.75, 1.0 | | |

Table D.4: Hyperparameter spaces for ML models in Reshuffling.

| ML model | Hyperparameter | Type | Range | Info |
|---|---|---|---|---|
| Funnel-Shaped MLP | learning rate | continuous | [1e-4, 1e-1] | log |
| | num layers | integer | [1, 5] | |
| | max units | categorical | 64, 128, 256, 512 | |
| | batch size | categorical. | 16, 32, ..., max_batch_size | |
| | momentum | continuous. | [0.1, 0.99] | |
| | alpha | continuous. | [1e-6, 1e-1] | log |
| Elastic Net | C | continuous | [1e-6, 10e4] | log |
| | l1 ratio | continuous | [0.0, 1.0] | |
| XGBoost | max depth | integer | [2, 12] | log |
| | alpha | continuous | [1e-8, 1.5] | log |
| | lambda | continuous | [1e-8, 1.0] | log |
| | eta | continuous | [0.01, 0.3] | log |
| CatBoost | learning rate | continuous | [0.01 ,0.3] | log |
| | depth | integer | [2, 12] | |
| | l2 leaf reg | continuous | [0.5, 30] | |

## D.3 Baselines and their hyperparameters

We use several bandit algorithms as baselines. Table D.5 summarizes the hyperparameters and their values.

Table D.5: Hyperparameters of Bandit Baselines.

| Algorithm | Hyperparameter | Value | Reference |
|---|---|---|---|
| MaxUCB | $\alpha$ | 0.5 | Ours |
| *Quantile Bayes UCB* | $\alpha$ $\beta$ $\tau$ | 1.0 0.2 0.95 | Balef et al. [2024] |
| *Quantile UCB* | $\alpha$ $\tau$ | 0.5 0.95 | |
| *ER-UCB-S* | $\beta$ $\theta$ $\gamma$ | 0.6 0.01 20.0 | Hu et al. [2021] |
| *ER-UCB-N* | $\alpha$ $\theta$ $\gamma$ | 1.0 0.01 20.0 | |
| *Rising Bandits* | $C$ $T$ | 7 Time horizon | Li et al. [2020] |
| *Max-Median* | $\epsilon$ | $1/(t)$, $t$ is iteration | Bhatt et al. [2022] |
| *QoMax-SDA* | $q$ $\gamma$ | 0.5 2/3 | Baudry et al. [2022] |
| *QoMax-ETC* | $q$ $b_T$ $n_T$ $T$ | 0.5 4 3 Time horizon | |
| *UCB* | $\alpha$ | 0.5 | Auer [2002] |
| *ThresholdAscent* | $\delta$ $s$ $T$ | 0.1 20 Time horizon | Streeter and Smith [2006b] |
| *Successive Halving* | $\eta$ $T$ | 2.0 Time horizon | Karnin et al. [2013] |
| *R-SR* | $\epsilon$ $T$ | 0.25 Time horizon | Mussi et al. [2024] |
| *R-UCBE* | $\alpha$ $\epsilon$ $\sigma$ $T$ | 57.12 0.25 0.05 Time horizon | |
| *MaxSearch (Gaussian)* | $c$ | 1.0 | Kikkawa and Ohno [2024] |
| *MaxSearch (SubGaussian)* | $c$ | 0.27 | |

**D.4 More Results on the Sensitivity Analysis of Hyperparameter $\alpha$**

In addition to the results shown in Figure 5 in Section 5, we provide additional results here. We evaluated the performance of MaxUCB for $\alpha \in [0, 2.9]$ with step size 0.1. We plot the performance over the number of iterations for different values of $\alpha$ in Figures D.1, D.2, D.3, D.4. Green indicates better performance, showing the impact of $\alpha$ at different stages of the optimization procedure. Furthermore, we provide a comparison between MaxUCB with different values of $\alpha$ with *combined search*(*SMAC*) in Table D.6. For the experiments in the main paper, we choose $\alpha = 0.5$ as a robust choice over all datasets. Notably, $\alpha = 0.5$ is selected based on the assumption that the reward distribution support is $[0, 1]$. For other supports, we recommend scaling $\alpha$ according to the range of the support.

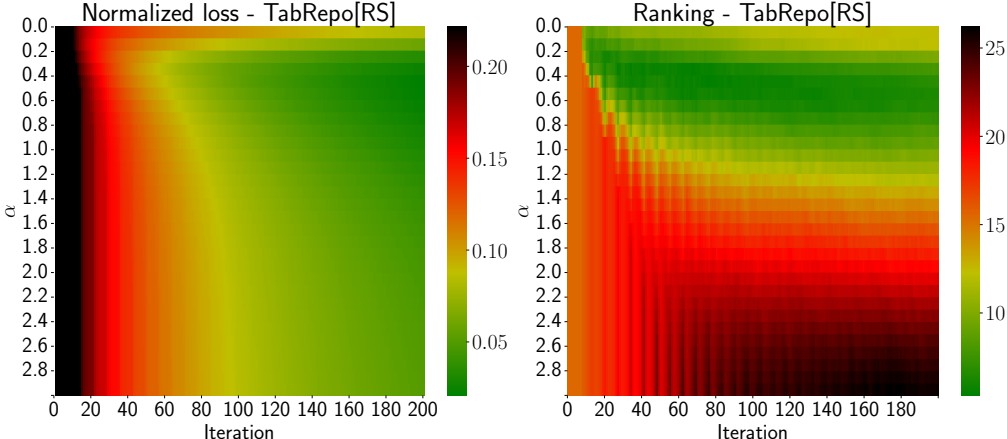

Figure D.1: Heatmap showing the performance of our algorithm with different values of $\alpha$ for TabRepo dataset.

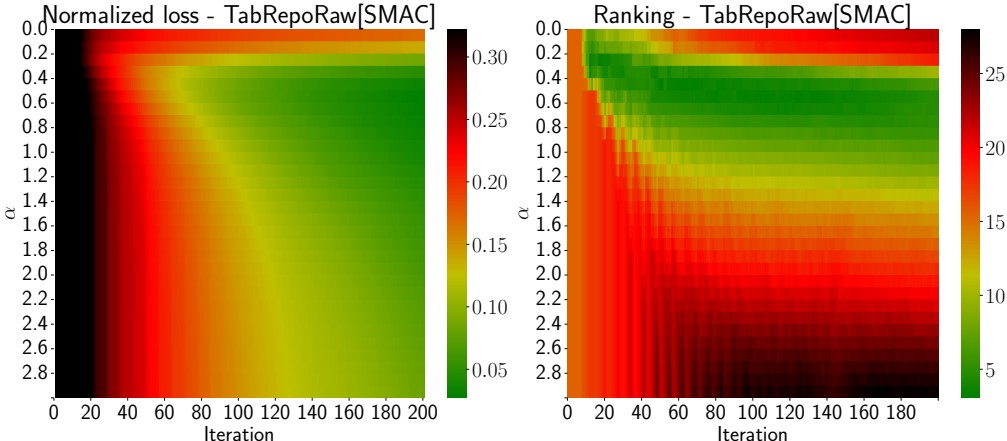

Figure D.2: Heatmap showing the performance of our algorithm with different values of $\alpha$ for TabRepoRaw dataset.

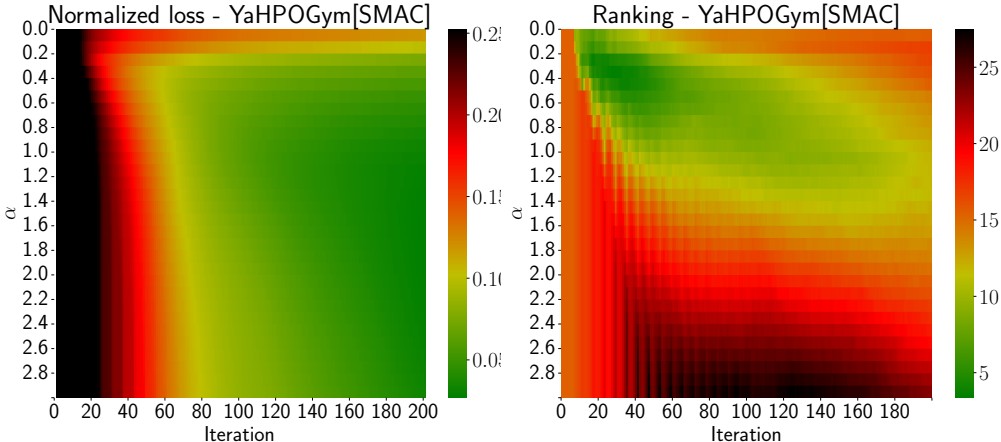

Figure D.3: Heatmap showing the performance of our algorithm with different values of $\alpha$ for YaHPOGym dataset.

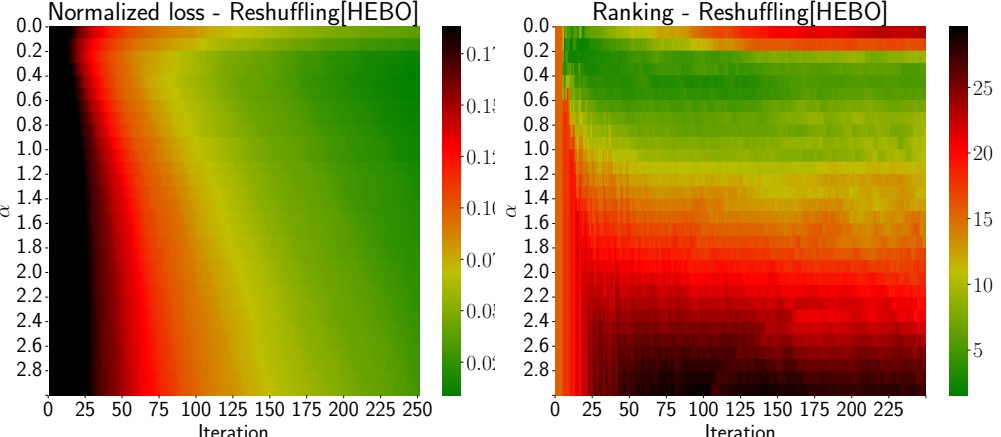

Figure D.4: Heatmap showing the performance of our algorithm with different values of $\alpha$ for Reshuffling dataset.

Table D.6: Comparing MaxUCB with *combined search*(*SMAC*) for different values of $\alpha$ and time steps. P-values from a sign test assessing whether bandit methods outperform *combined search*. P-values below $\alpha = 0.05$ are underlined, while those below $\alpha = 0.05$ after multiple comparison correction (adjusting $\alpha$ by the number of comparisons) are boldfaced and underlined indicating that the two-level approach is superior to *combined search*. Additionally, we report the normalized loss and the number of wins, ties, and losses (w/t/l) of bandit methods.

| | Time | | $\alpha$=0.0 | $\alpha$=0.1 | $\alpha$=0.2 | $\alpha$=0.3 | $\alpha$=0.4 | $\alpha$=0.5 | $\alpha$=0.6 | $\alpha$=0.7 | $\alpha$=0.8 | $\alpha$=0.9 |
|---|---|---|---|---|---|---|---|---|---|---|---|---|
| **TabRepoRaw[SMAC]** | 50 | p-value | 0.1002 | **0.0214** | **0.0000** | **0.0000** | **0.0000** | **0.0000** | **0.0000** | **0.0000** | **0.0000** | **0.0000** |
| | | w/t/l | 19/0/11 | 21/0/9 | 27/0/3 | 29/0/1 | 29/0/1 | 29/0/1 | 29/0/1 | 28/0/2 | 27/0/3 | 27/0/3 |
| | | loss | 0.2555 | 0.2446 | 0.2172 | 0.1946 | 0.1965 | 0.2134 | 0.2203 | 0.2229 | 0.2249 | 0.2296 |
| | 100 | p-value | 0.9974 | 0.9919 | 0.8998 | 0.1808 | **0.0081** | **0.0214** | 0.1002 | 0.5722 | 0.8192 | 0.9919 |
| | | w/t/l | 8/0/22 | 9/0/21 | 12/0/18 | 18/0/12 | 22/0/8 | 21/0/9 | 19/0/11 | 15/0/15 | 13/0/17 | 9/0/21 |
| | | loss | 0.2138 | 0.2053 | 0.1832 | 0.1346 | 0.1283 | 0.1113 | 0.1164 | 0.1208 | 0.1279 | 0.1512 |
| | 150 | p-value | 0.9993 | 0.9919 | 0.9506 | 0.2923 | **0.0007** | **0.0007** | **0.0026** | **0.0081** | 0.1002 | 0.7077 |
| | | w/t/l | 7/0/23 | 9/0/21 | 11/0/19 | 17/0/13 | 24/0/6 | 24/0/6 | 23/0/7 | 22/0/8 | 19/0/11 | 14/0/16 |
| | | loss | 0.1994 | 0.1911 | 0.1671 | 0.1020 | 0.0898 | 0.0752 | 0.0775 | 0.0849 | 0.0887 | 0.0973 |
| | 200 | p-value | 0.9998 | 0.9993 | 0.9786 | 0.1808 | **0.0214** | **0.0007** | **0.0026** | **0.0026** | **0.0081** | **0.0081** |
| | | w/t/l | 6/0/24 | 7/0/23 | 10/0/20 | 18/0/12 | 21/0/9 | 24/0/6 | 23/0/7 | 23/0/7 | 22/0/8 | 22/0/8 |
| | | loss | 0.1864 | 0.1790 | 0.1489 | 0.0686 | 0.0651 | 0.0563 | 0.0622 | 0.0698 | 0.0703 | 0.0751 |
| **YaHPOGym[SMAC]** | 50 | p-value | **0.0000** | **0.0000** | **0.0000** | **0.0000** | **0.0000** | **0.0000** | **0.0000** | **0.0000** | **0.0000** | **0.0000** |
| | | w/t/l | 74/1/28 | 83/1/19 | 95/1/7 | 102/1/0 | 102/1/0 | 102/1/0 | 102/1/0 | 102/1/0 | 102/1/0 | 101/1/1 |
| | | loss | 0.1853 | 0.1532 | 0.1071 | 0.0930 | 0.0942 | 0.0978 | 0.1047 | 0.1101 | 0.1151 | 0.1190 |
| | 100 | p-value | 0.3833 | 0.3833 | 0.3833 | **0.0459** | **0.0112** | **0.0112** | **0.0088** | 0.0572 | 0.2153 | 0.4220 |
| | | w/t/l | 53/1/49 | 53/1/49 | 53/1/49 | 60/1/42 | 63/1/39 | 63/1/39 | 64/0/39 | 60/0/43 | 56/0/47 | 53/0/50 |
| | | loss | 0.1494 | 0.1177 | 0.0876 | 0.0813 | 0.0782 | 0.0713 | 0.0668 | 0.0702 | 0.0718 | 0.0749 |
| | 150 | p-value | 0.3833 | 0.6167 | 0.8135 | 0.0686 | **0.0036** | **0.0065** | **0.0028** | **0.0005** | **0.0028** | **0.0148** |
| | | w/t/l | 53/1/49 | 50/1/52 | 47/1/55 | 59/1/43 | 65/1/37 | 64/1/38 | 66/0/37 | 68/1/34 | 66/0/37 | 63/0/40 |
| | | loss | 0.1378 | 0.1013 | 0.0758 | 0.0722 | 0.0716 | 0.0551 | 0.0518 | 0.0507 | 0.0520 | 0.0500 |
| | 200 | p-value | 0.5394 | 0.8135 | 0.6896 | 0.2442 | **0.0185** | **0.0088** | **0.0015** | **0.0002** | **0.0000** | **0.0001** |
| | | w/t/l | 51/1/51 | 47/1/55 | 49/1/53 | 55/1/47 | 62/1/40 | 64/0/39 | 67/0/36 | 70/0/33 | 75/0/28 | 71/0/32 |
| | | loss | 0.1190 | 0.0856 | 0.0697 | 0.0660 | 0.0614 | 0.0457 | 0.0443 | 0.0418 | 0.0423 | 0.0421 |

## D.5 More Results for the Empirical Evaluation

In addition to the analysis in the main paper in Figure 6 in Section 5, we report the averaged normalized loss over time in Figure D.5, the average ranking in Figure D.6, the normalized loss per task in Figure D.7, the ranking per task in Figure D.8 and critical distance plots as described by Demšar [2006] in Figure D.9. Additionally, we report results for a *Random Policy* (yellow) that selects arms to pull at random.

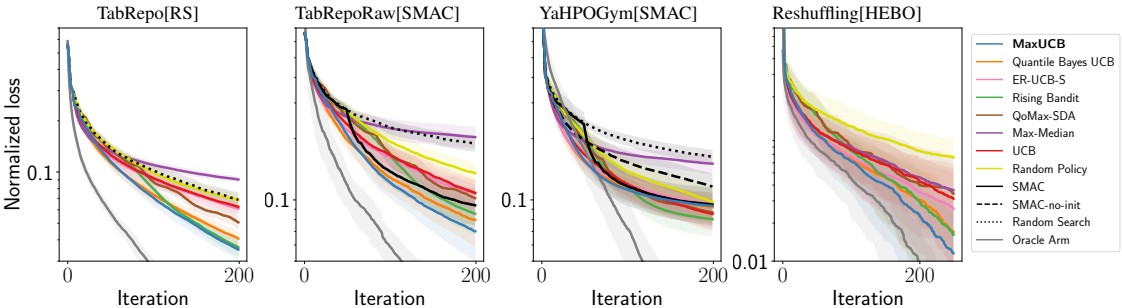

Figure D.5: Average normalized loss of algorithms on different benchmarks, lower is better. *SMAC* and *random search* perform *combined search* across the joint space.

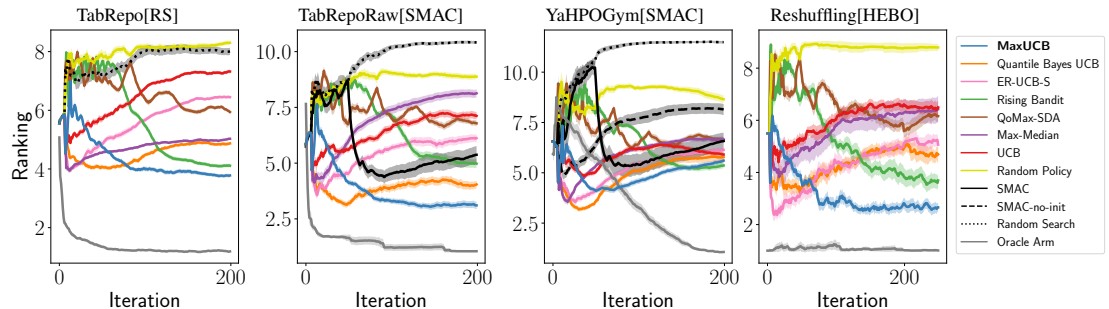

Figure D.6: Average ranking of algorithms on different benchmarks, lower is better. *SMAC* and *random search* perform *combined search* across the joint space.

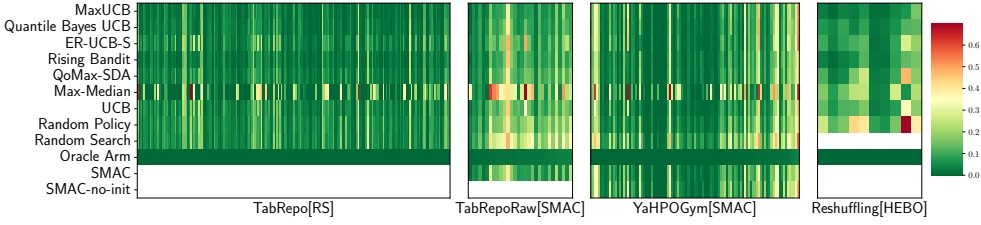

Figure D.7: Heatmap showing the normalized loss of algorithms per task of each benchmark, sorted by the oracle arm performance, lower is better. *SMAC* and *random search* perform *combined search* across the joint space.

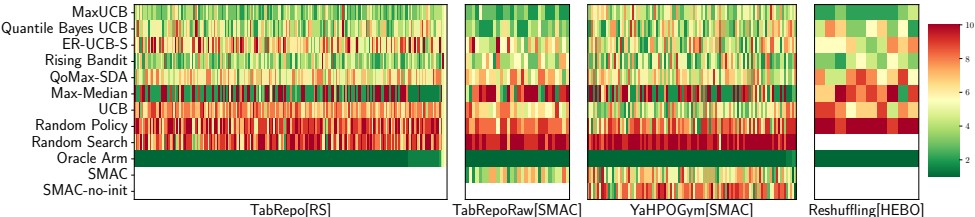

Figure D.8: Heatmap showing the ranking of algorithms per task of each benchmark, sorted by the oracle arm performance, lower is better. *SMAC* and *random search* perform *combined search* across the joint space.

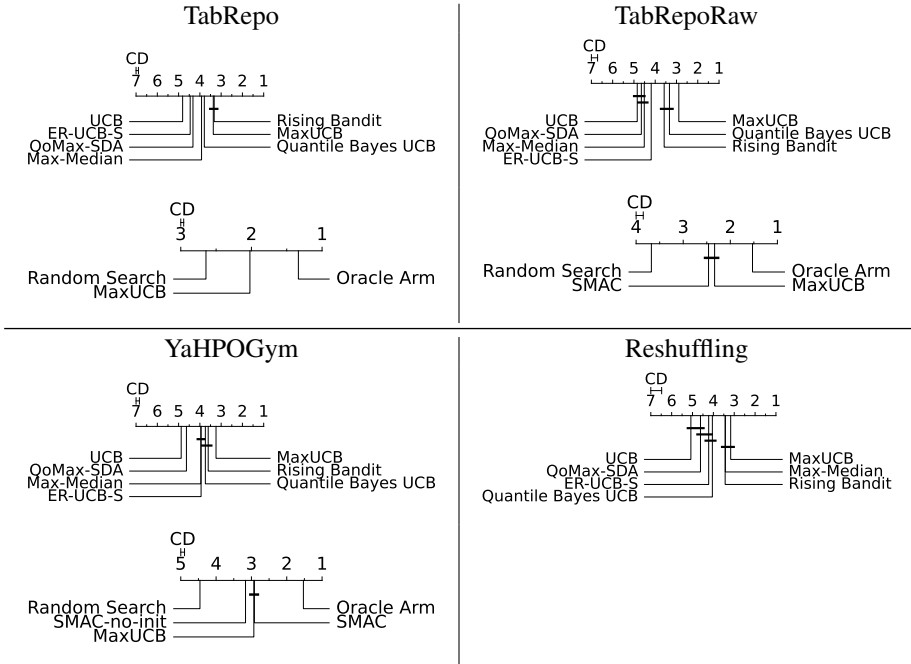

Figure D.9: Diagrams to compare the performance (ranking) of different algorithms using the Critical Distance (CD). For each benchmark on top, we compare bandit methods, and on the bottom, we compare MaxUCB against *combined search* and the oracle arm.

## D.6 More Baselines for the Empirical Evaluation

**Max $K$-armed Bandit Baselines.** We compare MaxUCB against *MaxSearch Gaussian* [Kikkawa and Ohno, 2024], *MaxSearch SubGaussian* [Kikkawa and Ohno, 2024], *QoMax-ETC* [Baudry et al., 2022], *QoMax-SDA* [Baudry et al., 2022], *Max-Median* [Bhatt et al., 2022] and *Threshold Ascent* [Streeter and Smith, 2006b]. We report the averaged normalized loss over time in Figure D.10, the average ranking in Figure D.11. As shown, our algorithm outperforms all extreme bandit algorithms in these benchmarks.

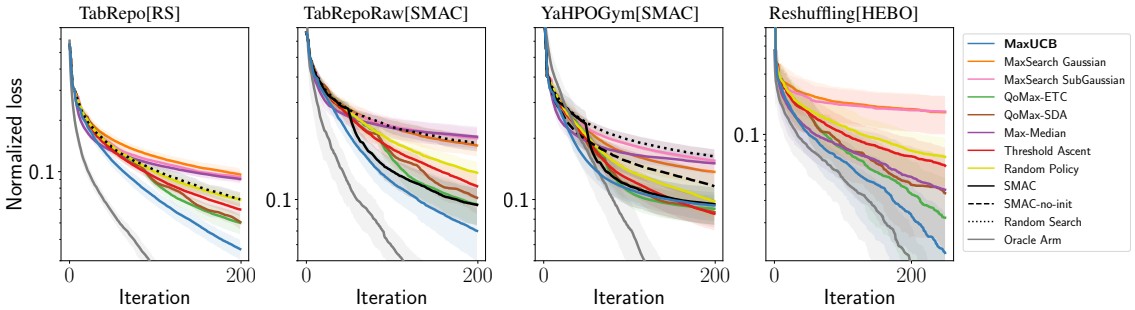

Figure D.10: Average normalized loss of MKB algorithms on different benchmarks, lower is better. *SMAC* and *random search* perform *combined search* across the joint space.

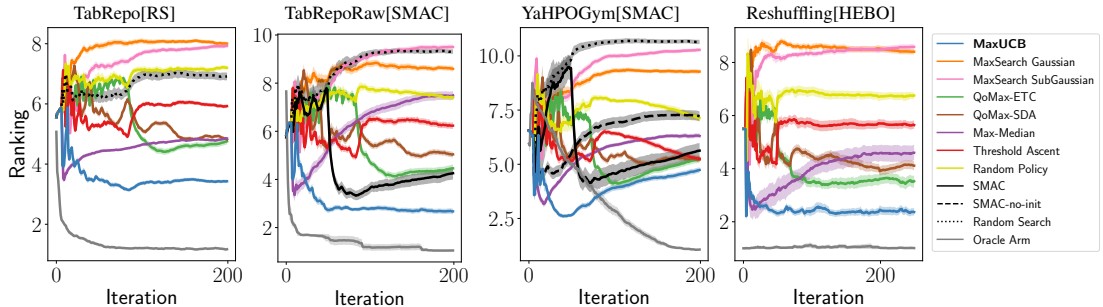

Figure D.11: Average ranking of MKB algorithms on different benchmarks, lower is better. *SMAC* and *random search* perform *combined search* across the joint space.

**A Few More Relevant Bandit Baselines.** We compare MaxUCB against *Quantile UCB* [Balef et al., 2024], *ER-UCB-N* [Hu et al., 2021], *R-SR* [Mussi et al., 2024], *R-UCBE* [Mussi et al., 2024], *Successive Halving* [Karnin et al., 2013] and *EXP3* [Auer et al., 2002]. We report the averaged normalized loss over time in Figure D.12, the average ranking in Figure D.13.

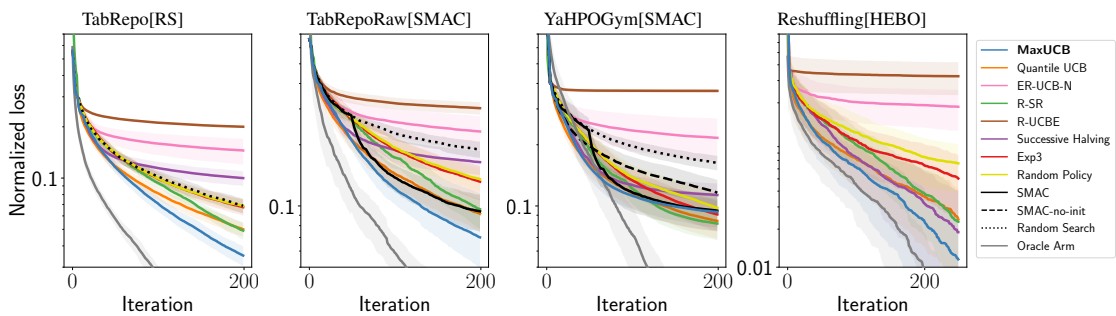

Figure D.12: Average normalized loss of algorithms on different benchmarks, lower is better. *SMAC* and *random search* perform *combined search* across the joint space.

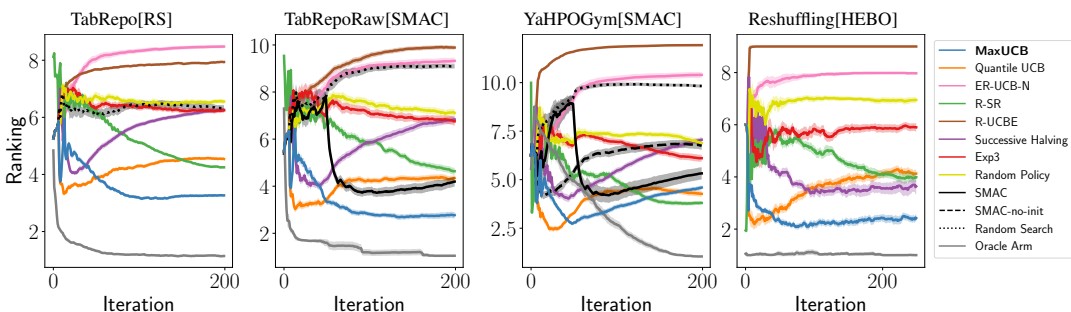

Figure D.13: Average ranking of algorithms on different benchmarks, lower is better. *SMAC* and *random search* perform *combined search* across the joint space.

# E  More Details on the Empirical Behaviour of MaxUCB

Here, we provide further analysis of MaxUCB. Concretely, we study how often our algorithm pulls the optimal arm and compare it to the theoretical results. Furthermore, we evaluate an extension of MaxUCB to handle non-stationary rewards and finally study MaxUCB performance on common synthetic benchmarks used in the extreme bandit literature.

## E.1  The number of times each arm is pulled

Proposition 4.1 shows that the number of times the optimal arm is pulled can be viewed as a good metric for measuring the performance of algorithms. Figure E.1, E.2,E.3,E.4 shows the average number of pulling arms on different benchmarks. They indicate that, on average, MaxUCB, *Rising Bandits*, and *Max-Median* algorithms often choose the optimal arm. However, for *Max-Median*, the number of pulls of the optimal arm is either very close to $0$ or to $T$, leading to a non-robust performance, which has already been observed in Baudry et al. [2022] experiments. *UCB* and *ER-UCB-S* perform almost similarly.

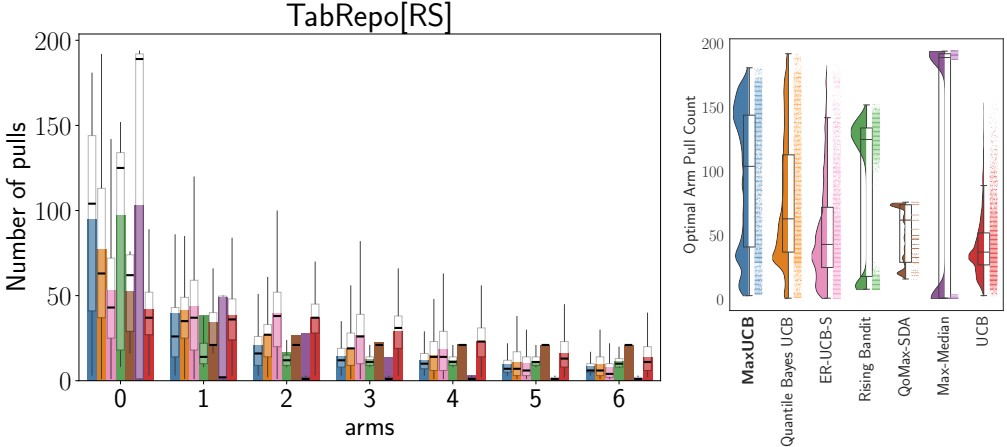

Figure E.1: (Right) The number of all arm pulls, with each bar graph showing the average and the error bars indicating additional statistical information. (Left) The number of best arm pulls for different bandit algorithms.

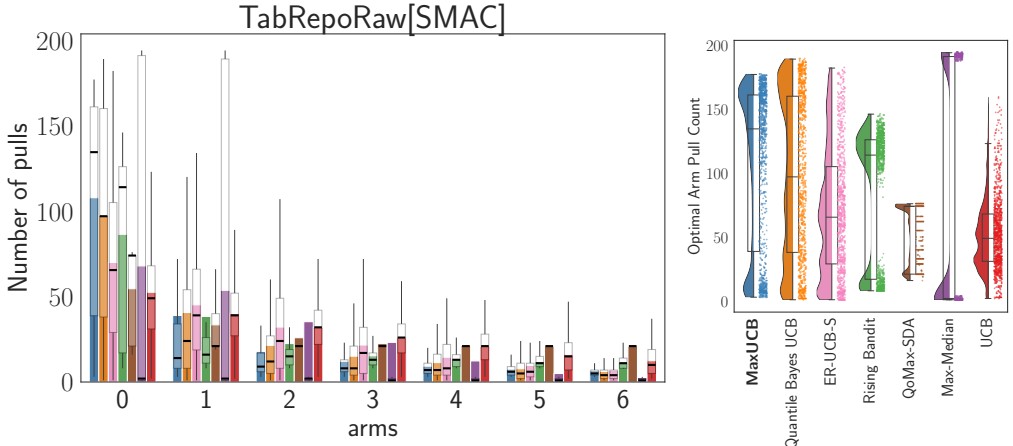

Figure E.2: (Right) The number of all arm pulls, with each bar graph showing the average and the error bars indicating additional statistical information. (Left) The number of best arm pulls for different bandit algorithms.

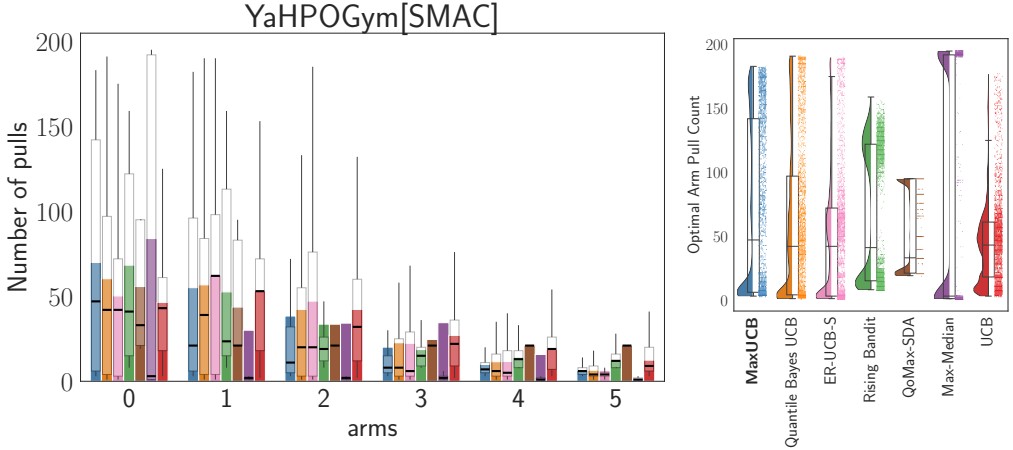

Figure E.3: (Right) The number of all arm pulls, with each bar graph showing the average and the error bars indicating additional statistical information. (Left) The number of best arm pulls for different bandit algorithms.

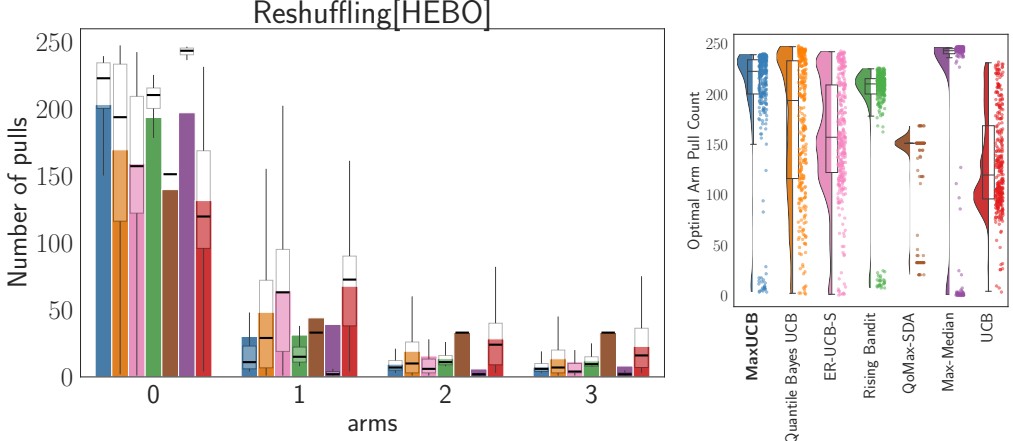

Figure E.4: (Right) The number of all arm pulls, with each bar graph showing the average and the error bars indicating additional statistical information. (Left) The number of best arm pulls for different bandit algorithms.

## E.2 From theory to practice

To validate our theorem against practical outcomes, we applied our algorithm to all benchmarks and plotted the number of pulls for each arm, denoted as "Real Experiment." Additionally, we computed the upper bound on the number of pulls by using the empirical values of $L_1$ and $U_i$ and $\Delta_i$. Notably, we report the first term of Equation 22 since the second term is nearly constant across all arms according to calculation. The results demonstrate that although the empirical pull counts are much less than the theoretical bounds of Equation 22, both follow a similar decreasing pattern as the rank of suboptimality increases.

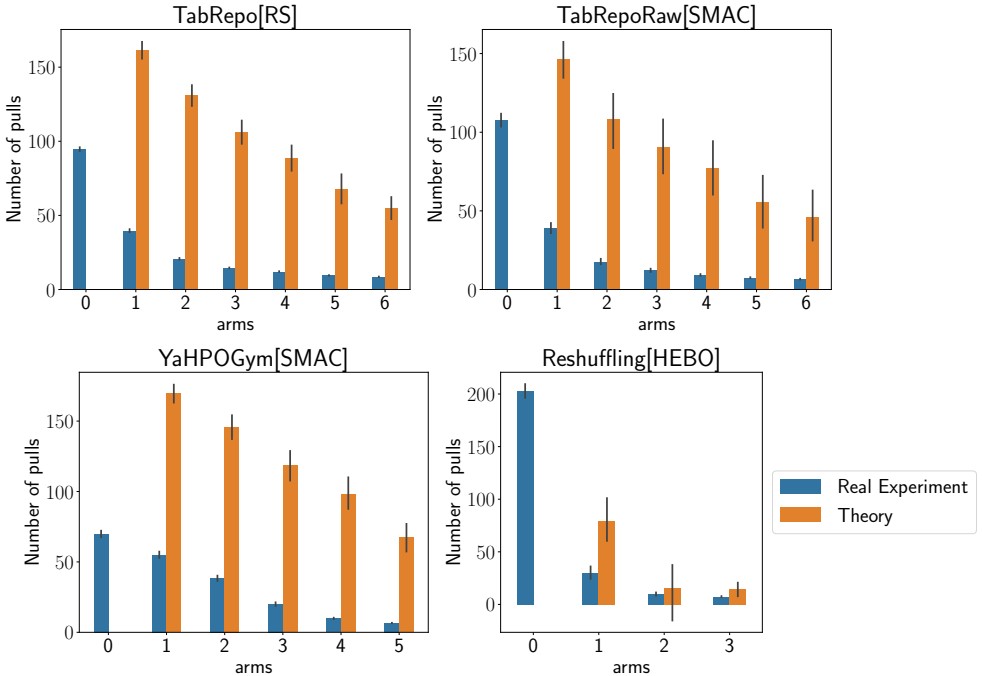

Figure E.5: The number of pulls for each arm in our algorithm, labeled as "Real Experiment" and the theoretical values of this number

## E.3 Addressing Non-stationary Rewards

To handle non-stationary rewards, we pull each arm $C$ times without observing the rewards before running MaxUCB. This "burn-in" allows the Markov Chain to reach equilibrium, especially from a poor starting point. Algorithm E.1 shows the adapted version of our algorithm. Therefore, empirically, this allows MaxUCB to operate after a fixed exploration phase of all arms until the reward distribution is stationary.

We run Algorithm E.1 with different parameters of $C \in \{5, 6, 7, 8\}$ using up to 48 iterations corresponding to almost 25% of the total budget. Figure E.6 shows normalized loss per task where columns are sorted by the maximum change of the mean of the reward distributions of the optimal arm computed every 10 HPO iterations (as an indicator of non-stationarity; shown at the top panel in Figure E.6. Figure E.7 shows the average ranking and normalized loss over time for different values of the hyperparameter $C$.

The initial burn-in improves final performance for the few tasks where we observe a high shift (right part of Figure E.6) while the initial performance is worse across all tasks (as shown in Figure E.7). However, the results are not sensitive to the exact value of $C$. Overall, this naive solution can improve performance for some tasks at the cost of not using potentially valuable information obtained from initial exploration. Thus, optimally addressing non-stationary rewards could be a promising direction for future work.

**Algorithm E.1** MaxUCB-Burn-in

**Require:** $\alpha$(exploration parameter), $C$ (burn-in rounds) , $T$(time horizon), $K$(arms) ▷ Burn-in phase
1: **for** $j \leq C$, for each arm $i \leq K$ **do**
2:     Pull arm $i$
3: **end for** ▷ Initial phase
4: **for** each arm $i \leq K$ **do**
5:     Pull arm $i$
6:     set $n_i \leftarrow 1$, observe reward $r_{i,1}$
7: **end for** ▷ Main phase
8: **for** $t = (CK + K + 1)$ to $T$ **do**
9:     **for** each arm $i \leq K$ **do**
10:         Update policy $U_i = \max(r_{i,1}, ..., r_{i,n_i}) + (\frac{\alpha \log(t)}{n_i})^2$
11:     **end for**
12:     Select arm $I_t = \underset{i \leq K}{\arg\max}\, U_i$
13:     $n_{I_t} \leftarrow n_{I_t} + 1$
14:     Observe reward $r_{I_t, n_{I_t}}$
15: **end for**

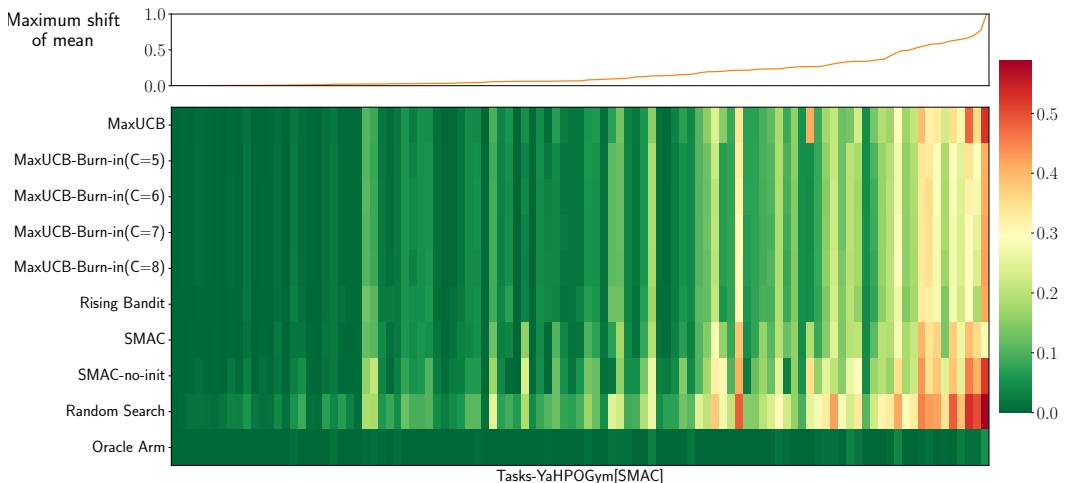

Figure E.6: Heat map shows normalized loss per task, sorted by the distribution shift.

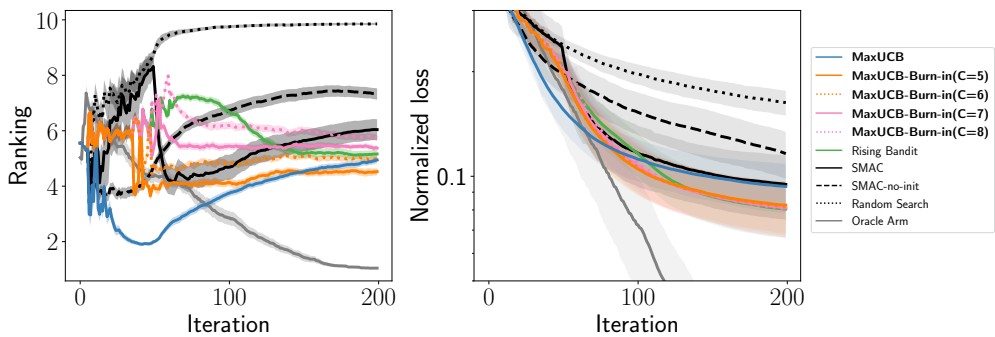

Figure E.7: Average rank and normalized loss of algorithms on YaHPOGym benchmark, lower is better.

### E.4 Toy examples from the extreme bandit's literature

In this section, we provide additional results on commonly used benchmark functions used in the extreme bandit literature. Concretely, we use a similar setup to [Baudry et al., 2022] and report the following four tasks:

1. $K = 5$ Pareto distributions with tail parameters $\lambda_k = [2.1, 2.3, 1.3, 1.1, 1.9]$. Results are shown in Figure E.8.

2. $K = 10$ Exponential arms with a survival function $G_k(x) = e^{-\lambda_k x}$ with parameters $\lambda_k = [2.1, 2.4, 1.9, 1.3, 1.1, 2.9, 1.5, 2.2, 2.6, 1.4]$. Results are shown in Figure E.9.

3. $K = 20$ Gaussian arms, with same mean $\mu_k = 1, \forall k$, and different variances $\sigma_k = [1.64, 2.29, 1.79, 2.67, 1.70, 1.36, 1.90, 2.19, 0.80, 0.12, 1.65, 1.19, 1.88, 0.89, 3.35, 1.5, 2.22, 3.03, 1.08, 0.48]$. The dominant arm has a standard deviation of $3.35$. Results are shown in Figure E.10.

4. (our toy example) $K = 5$ power distributions with domain parameter $[3, 4, 5, 5, 4]$ and shape parameter $[1.01, 1.01, 1.01, 1.1, 1]$. Results are shown in Figure E.11.

For each task, we run $N = 1000$ independent repetitions for six time horizons $T \in \{50, 100, 200, 500, 1000, 2000\}$. We show the CDF of the rewards for each arm, the number of times the optimal arm was pulled, and the proxy empirical regret. Notably, the proxy empirical regret is introduced by Baudry et al. [2022] to overcome the issue of high variance in the maximum values of distributions.

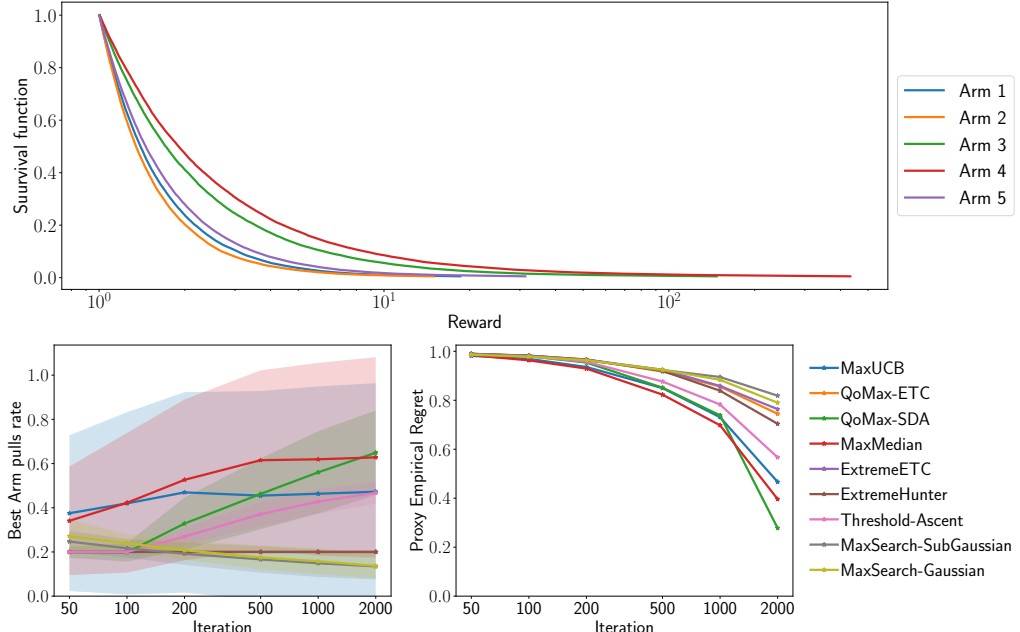

Figure E.8: Experiment 1: (Top) Survival function of distribution of each arm (left) Number of pulls of the optimal arm. (Right) Proxy Empirical Regret

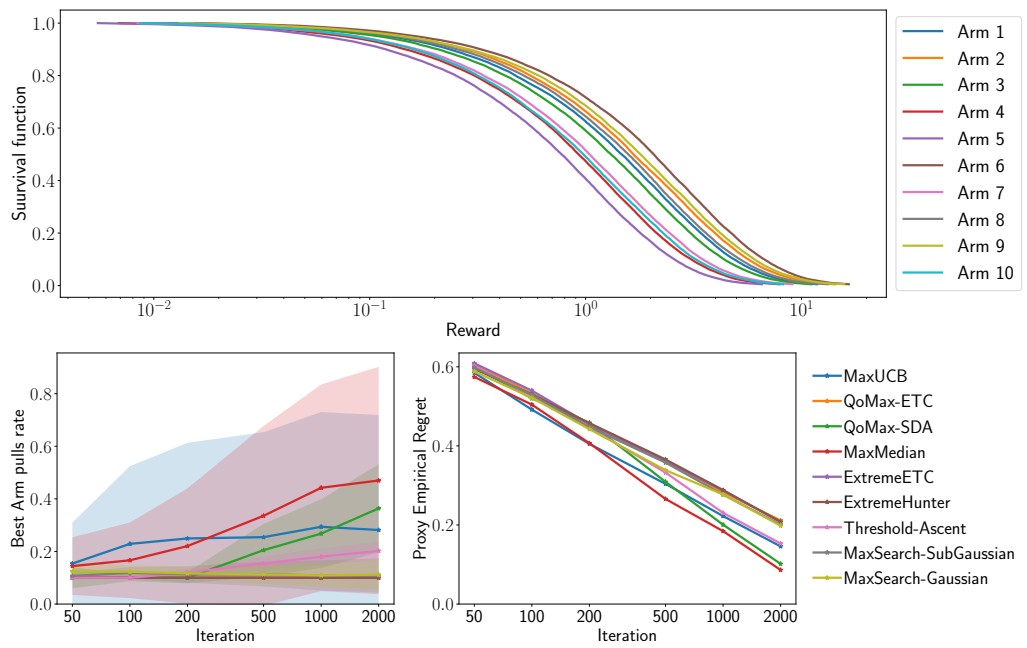

Figure E.9: Experiment 2: (Top) Survival function of distribution of each arm (left) Number of pulls of the optimal arm. (Right) Proxy Empirical Regret

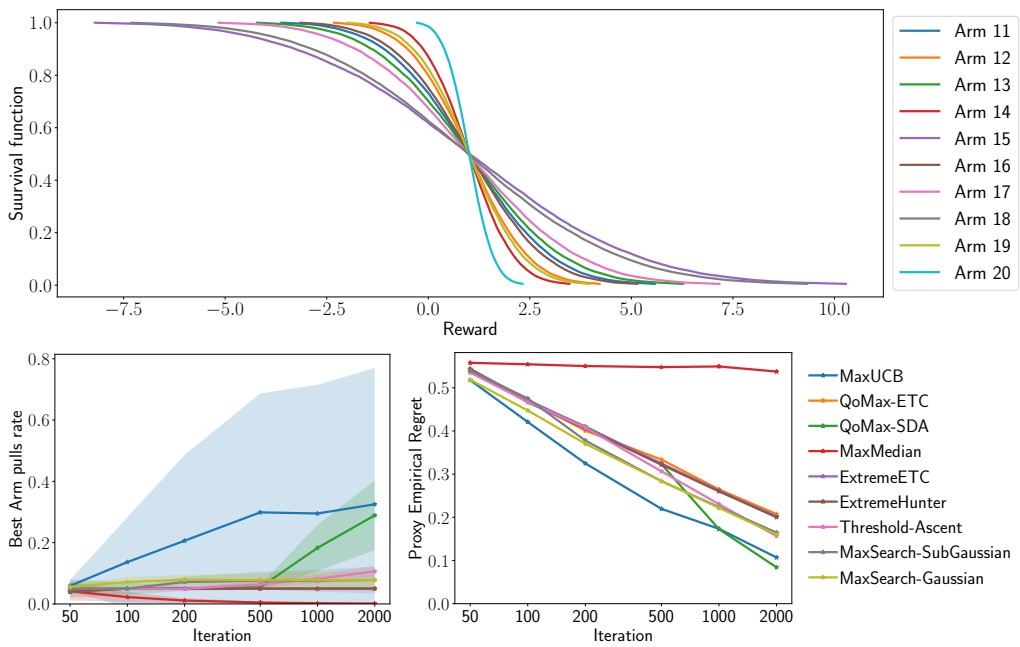

Figure E.10: Experiment 3: (Top) Survival function of distribution of some arms (left) Number of pulls of the optimal arm. (Right) Proxy Empirical Regret

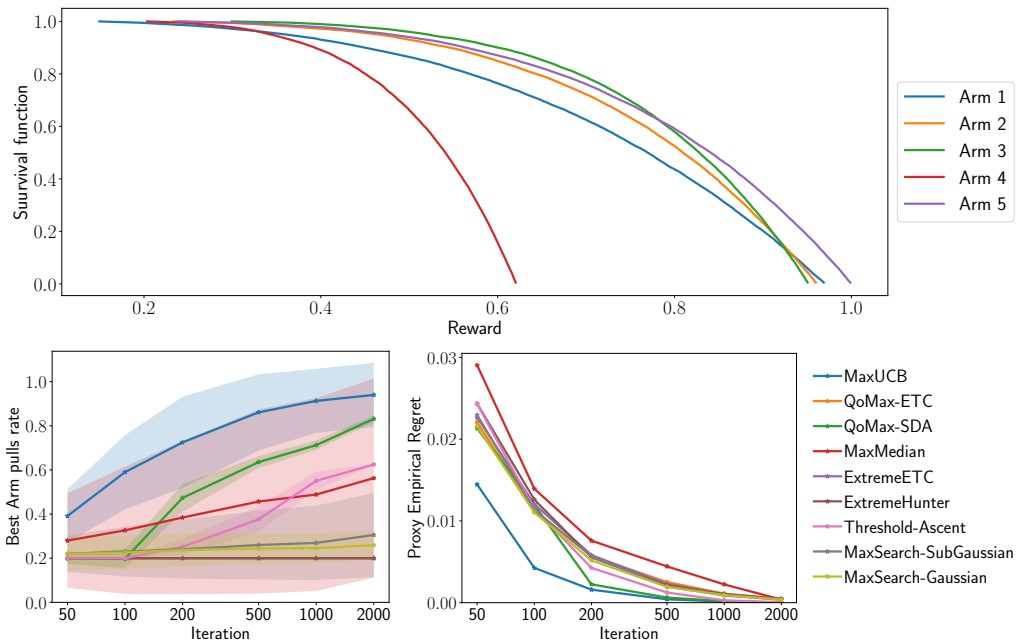

Figure E.11: Experiment 4: (Top) Survival function of distribution of each arm (left) Number of pulls of the optimal arm. (Right) Proxy Empirical Regret

## E.5 Supernet selection in Few-Shot Neural Architecture Search

In one-shot NAS, a single supernet approximates all architectures. However, this estimation could be inaccurate. To address this, few-shot NAS splits the supernet into smaller sub-supernets [Hu et al., 2022, Ly-Manson et al., 2024]. To further improve performance, [Hu et al., 2022] introduced supernet selection, which identifies the most promising sub-supernet and its optimal architecture using techniques such as *Successive Halving* . We aim to identify the best-performing architecture among several subspaces, each corresponding to a sub-supernet. This problem is analogous to the MKB problems, where each arm represents a sub-supernet.

**Dataset Preparation.** We use data provided in [Ly-Manson et al., 2024] for three benchmark datasets: *CIFAR-10*, *CIFAR-100*, and *ImageNet16-120*. Each dataset's search space is split using 10 different metrics. The splitting follows a binary tree structure with a depth of 3, where operations are divided into two groups at each branch. This process results in 8 sub-supernets per metric. Consequently, for each dataset, we have one full search space and 8 sub-search spaces.

Each combination of dataset and splitting metric is treated as a separate task, yielding a total of 30 tasks (3 datasets × 10 metrics), each with 8 arms. Following the setup of [Ly-Manson et al., 2024], we randomly sample 600 architectures from both the full search space and each sub-search space. This process is repeated 32 times using different random seeds to ensure variability and robustness in the results.

**Analyzing the reward distribution.** Figure E.12 illustrates the empirical survival functions of the rewards and sub-optimality gaps for the benchmark. As shown, the distribution shape is similar to that of HPO tasks: both are bounded and left-skewed. However, the sub-optimality gap is considerably smaller than HPO tasks, suggesting that identifying the optimal arm is more challenging and may require additional iterations. In Figure E.13, we show values of $L$ and $U$ from Lemma 3.3 for this benchmark.

**Performance Analysis.** Figure E.14 presents the average ranking and normalized loss of various bandit algorithms in this benchmark. As shown, *Successive Halving*, *Max-Median*, and *ER-UCB-S* perform well with a small time budget but fail to explore sufficiently to identify the optimal arm. *Rising Bandits*, as a fixed-confidence best-arm identification method, struggles to find the optimal arm. In contrast, MaxUCB outperforms all other baselines, demonstrating better performance when searching the entire search space with a higher budget.

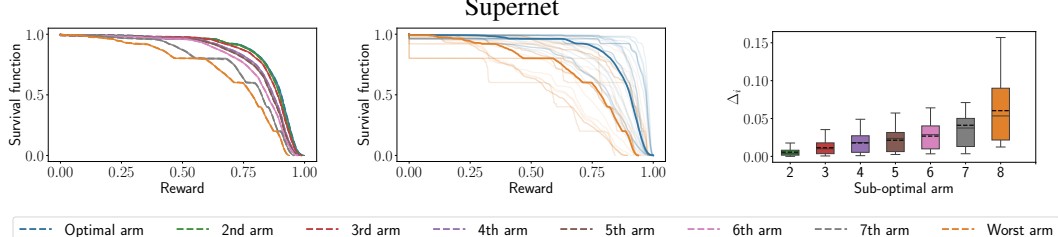

Figure E.12: (Left) The average empirical survival function of the reward (observed performances) per arm ranked per dataset. We divided the reward sequence into five segments over the budget to show the distribution change over time. Thin lines correspond to the survival function of different segments, visualizing the change over time. (Middle) The average ECDF per dataset for the best and worst arm with thin lines corresponding to individual datasets. (Right) The sub-optimality gap $\Delta_i$.

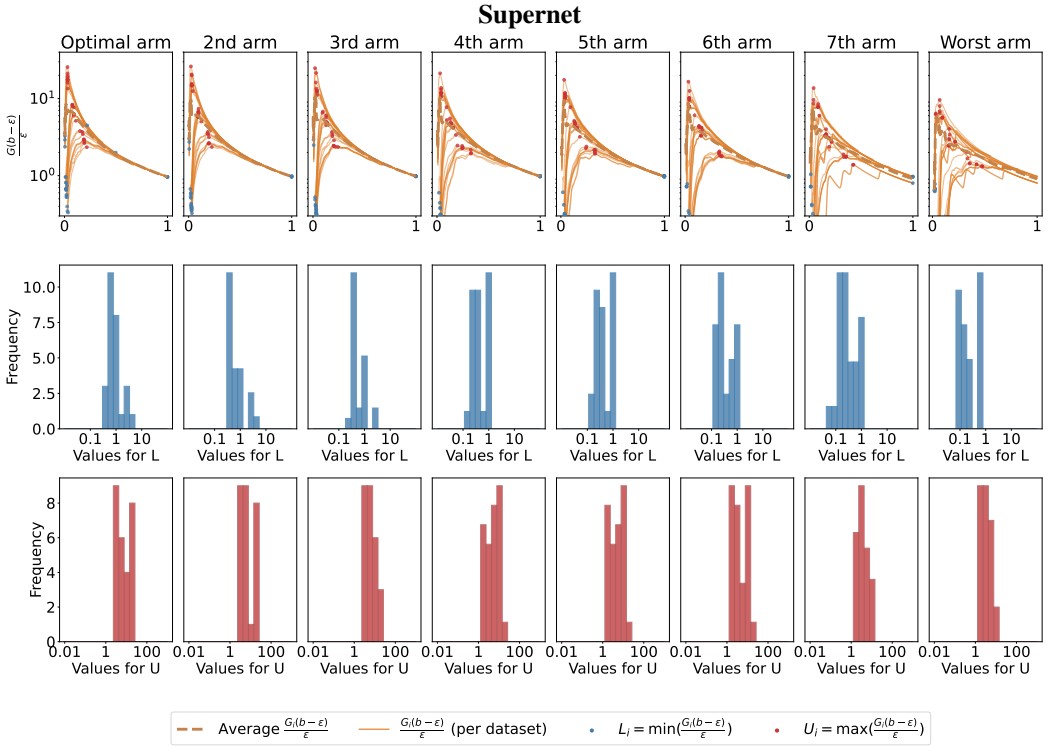

Figure E.13: Arms are ordered by sub-optimality gap. (Top) Thin orange lines represent $\frac{G(b-\epsilon)}{\epsilon}$, while the blue and red points correspond to $L$ and $U$ for our empirical reward distributions (see Lemma 3.3 for details). (Middle) Histogram of values for $L$. (Bottom) Histogram of values for $U$.

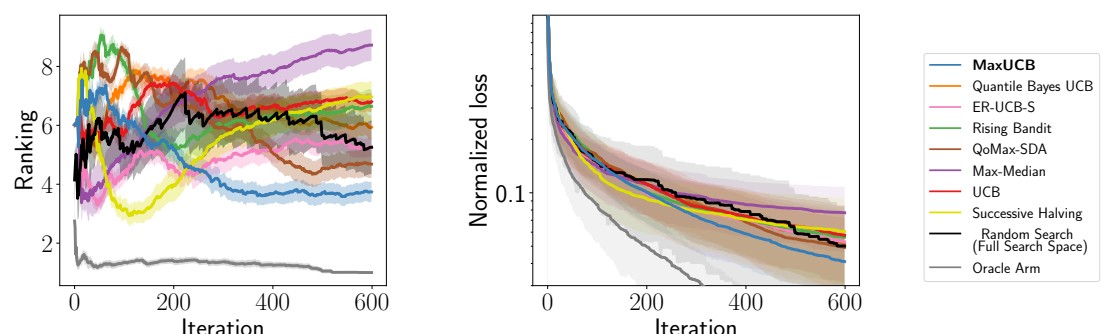

Figure E.14: Average rank and normalized loss of algorithms on *Supernet Selection* benchmark, lower is better.

