# OpenReview forum: "Put CASH on Bandits: A Max K-Armed Problem for Automated Machine Learning"
_NeurIPS.cc/2025/Conference — NeurIPS 2025 poster_

### Official Review · Reviewer_PkJH · 2025-06-29

**Clarity:** 4
**Significance:** 2
**Originality:** 3
**Rating:** 4
**Confidence:** 3

**Summary:**

This paper proposes MaxUCB, a new max k-armed bandit algorithm designed to address CASH problem in AutoML. The authors analyze empirical performance distributions from AutoML benchmarks and find them to be left-skewed and short-tailed. Based on this observation, they derive a regret bound and develop MaxUCB, which uses a tailored exploration bonus that accounts for the bounded distribution shape.

The method is evaluated on AutoML benchmarks and shows superior anytime and final performance compared to several state-of-the-art baselines, including other bandit-based algorithms and combined search approaches. The paper also provides an extensive theoretical analysis and discusses the sensitivity of the algorithm’s hyperparameter $\alpha$.

**Questions:**

Could the authors clarify how the tabular datasets used in the empirical distribution analysis were selected? For example, were these chosen to be representative of a specific class of AutoML problems, or were they selected based on convenience or availability?

**Ethical Concerns:**

["NO or VERY MINOR ethics concerns only"]

**Limitations:**

While the paper frames MaxUCB as a novel contribution, the core idea using a UCB-style bandit strategy with an exploration bonus adapted to bounded reward distributions can be seen as a relatively straightforward extension of standard UCB principles.

**Quality:**

2

**Strengths And Weaknesses:**

Strengths


The paper is logically organized, with a clear progression from problem formulation to method design, theoretical results, and empirical evaluation. The authors do a good job contextualizing prior work and explaining.


The combination of (i) empirical analysis of reward distributions, (ii) a tailored regret bound under bounded, left-skewed distributions, and (iii) the design of MaxUCB is a meaningful over prior work.


The paper includes thorough empirical comparisons across multiple benchmarks, careful analysis of the $\alpha$ parameter, and ablation against several strong baselines.


Weaknesses


While the authors analyze reward distributions to motivate their assumptions, there is no clearly defined methodology for how the tabular datasets were selected, nor a principled justification that the chosen benchmarks are representative of the broader diversity of AutoML tasks. This raises questions about whether the conclusions about bounded, short-tailed distributions hold more generally.

---

> ### Author Rebuttal · Authors · 2025-07-30
>
> Thanks a lot for the detailed review and the raised points. We particularly appreciate your highlighting the quality of explanation and clarity.
>
> ---
> ### “Could the authors clarify how the tabular datasets used in the empirical distribution analysis were selected? For example, were these chosen to be representative of a specific class of AutoML problems, or were they selected based on convenience or availability?”
> We used several well-established and widely used benchmark sets that were developed to compare HPO methods. Each benchmark contained different datasets, tasks, and search spaces, to ensure our empirical distribution analysis was not limited to a single source or problem type.
>
> * For example, **TabRepo** is one of the largest tabular benchmarks, including **200 classification and regression datasets** and evaluating over 1,310 models and configurations. A full evaluation of TabRepo could require up to 4,066,576 CPU hours.
>
> * For **TabRepoRaw**, where we ran Bayesian optimization directly over TabRepo’s search space, we selected the D244_F3_C1530_30 subset due to cost limitations. This subset includes **30 datasets** that were specifically chosen by the TabRepo authors to represent a diverse yet computationally feasible set of tasks. We spent over 10,000 CPU hours to collect this benchmark.
>
> * **YaHPOgym** is a well-known and widely used surrogate benchmark for HPO in the tabular domain, containing 103 datasets from the OpenML collection.
>
> * We also added the **Reshuffling** benchmark which includes 10 datasets with and HEBO as the HPO method. We included this task to diversify the HPO methods in our study.
>
> * Furthermore, in Appendix E.5, we used Supernet selection in the Few-Shot Neural Architecture Search benchmark (containing CIFAR-10, CIFAR-100, and ImageNet16-120 datasets) to show that our analysis and algorithm can work beyond CASH tasks.
>
> ---
> ### “While the paper frames MaxUCB as a novel contribution, the core idea using a UCB-style bandit strategy with an exploration bonus adapted to bounded reward distributions can be seen as a relatively straightforward extension of standard UCB principles.”
>
> The main novelty in MaxUCB is a distribution-adapted exploration bonus (which is a distinct approach compared to standard UCB literature). Our main contribution, and what makes the method work ultimately, is to identify a semi-parametric modelling of the distribution of the maxima we aim to optimize.
>
> The main difficulty is getting the details right in the modelling assumptions. There may be alternative, smarter, and more complex algorithmic designs for this problem, but **isn’t it remarkable that a simple algorithm achieves such good empirical performance on such a widely studied problem?**

---

> > ### Comment · Reviewer_PkJH · 2025-08-08
> >
> > Thanks to the author for the response. I'm keeping my score unchanged.

---

### Official Review · Reviewer_aB58 · 2025-06-30

**Clarity:** 3
**Significance:** 3
**Originality:** 3
**Rating:** 5
**Confidence:** 3

**Summary:**

This paper tackles the Combined Algorithm Selection and Hyperparameter Optimization (CASH) problem, a core challenge in Automated Machine Learning (AutoML). The authors propose MaxUCB, a novel max k-armed bandit algorithm that efficiently balances the exploration of model classes with hyperparameter optimization (HPO). A key insight of the paper is a thorough empirical analysis of reward distributions from real-world HPO tasks, showing that these are typically bounded, short-tailed, left-skewed, and nearly stationary—in contrast to assumptions underlying classical max bandit methods. Leveraging this, the authors design MaxUCB with a distribution-aware exploration bonus, supported by theoretical regret analysis and empirical validation on standard AutoML benchmarks. The proposed approach achieves strong performance improvements over existing methods.

**Questions:**

Please discuss or comment on the above weakness parts.

**Ethical Concerns:**

["NO or VERY MINOR ethics concerns only"]

**Final Justification:**

I believe this paper is solid and is valuable for advancing the field of AutoML, so I vote for acceptance for the current paper.

**Limitations:**

Yes.

**Paper Formatting Concerns:**

No.

**Quality:**

4

**Strengths And Weaknesses:**

**Strengths:**

- **Addresses a Timely and Practical Problem:** The paper focuses on the CASH problem, which is fundamental to making ML more accessible by automating the joint selection of algorithms and hyperparameters. The authors clearly articulate the **inefficiencies of traditional HPO** in high-dimensional, hierarchical spaces and the limitations of naively treating each model class independently.
- **Data-Driven Algorithm Design:** A standout contribution is the **empirical characterization of reward distributions** in HPO tasks. The authors show that these are **light-tailed, bounded, and nearly stationary**, and leverage these properties to **motivate a new exploration strategy**. This empirical grounding provides a solid justification for deviating from the classical heavy-tailed assumptions in prior work and **addresses a gap in understanding realistic bandit reward distributions in AutoML**.
- **Strong Theoretical and Algorithmic Contributions:** The paper provides a well-structured theoretical analysis, including a **new sub-optimality gap definition** and a **distributional assumption (Assumption 3.2)** that is empirically supported. Lemma 3.3 further refines the distributional characterization near the maximum. The derived **regret bound (Proposition 4.1)** and **suboptimal pull bound (Theorem 4.2)** are tightly aligned with empirical observations, and the use of a **squared exploration bonus** $(\alpha \log(t)/n)^2$ is well-justified.
- **Empirical Performance:** MaxUCB shows **consistent gains in both anytime and final performance** across four well-established AutoML benchmarks (TabRepo, TabRepoRaw, YaHPOGym, and Reshuffling). The **two-level optimization framework using MaxUCB** significantly outperforms joint-space HPO methods, especially under tight iteration budgets.

---

**Weaknesses and Suggestions for Improvement:**

- **Assumptions on Reward Distributions and Stationarity:** While the assumptions of **boundedness and light-tailedness** are well-supported by data, the **near-stationarity assumption** may be restrictive. The paper notes that stationarity might not hold during early iterations of HPO, suggesting the need for a **burn-in phase**. Appendix E.3 explores this empirically, but a more **principled treatment of non-stationarity**, either algorithmically or theoretically, would enhance robustness and broaden applicability.
- **Lack of Lower Bound Analysis:** The paper does not provide a **lower bound on regret**, which leaves open the question of **whether MaxUCB is minimax optimal** in the bounded, light-tailed setting. Including such a result, or at least a discussion of potential optimality gaps, would strengthen the theoretical contributions.

---

> ### Author Rebuttal · Authors · 2025-07-30
>
> Thanks a lot for the detailed positive assessment of our work. We are happy to comment on the pointed-out suggestions for improvement.
>
> ---
> ### “Regarding the near-stationarity assumption,”
> As noted in our response to Reviewer 7Zgt, we acknowledge that this simplifying assumption may not strictly hold, but it provides a solid foundation for developing a strong algorithm. In classic bandits, non-stationarity is typically addressed by combining algorithms with i.i.d. assumptions with techniques such as sliding windows, discounting, or change-point detection, which have shown promising results in various applications. Since the max k-armed bandit has a different regret definition,  it may require specialized techniques. Designing such techniques is a promising research direction.
>
>
> ---
> ### “The paper does not provide a lower bound on regret,”
> Thanks a lot for raising this point. The notion of optimality for max K-armed bandits is unclear, as Nishihara et al. (2016) [1] pointed out. Namely, as opposed to classical bandit problems, **the best arm, and thus, the optimal policy, depends on the time horizon, which is here our compute budget.** They argue that the existence of no-regret algorithms fully depends on the distributional assumptions and show impossibility results in the Bernoulli case. A notion of optimality, if it exists, would have to be fully distribution-dependent, but for which distribution family? Our study focuses on CASH and proposes a useful set of assumptions on useful distributions, but we do not claim, nor aim to be optimal, since this notion remains elusive for this problem. Bounded extreme bandits are not extensively studied, potentially because use cases were lacking, but our work may open avenues for more fundamental research questions. We will clarify it in the paper.
>
> ---
> ###  References
> 1) Nishihara et al.: "No regret bound for extreme bandits." Artificial Intelligence and Statistics. PMLR, 2016.

---

### Official Review · Reviewer_Vu3f · 2025-07-01

**Clarity:** 3
**Significance:** 3
**Originality:** 3
**Rating:** 4
**Confidence:** 3

**Summary:**

This paper studies a challenging resource allocation problem in AutoML, i.e., Combined Algorithm Selection and Hyperparameter optimization (CASH). The authors propose MaxUCB, a max k-armed bandit method to trade off exploring different model classes and conducting hyperparameter optimization. MaxUCB is specifically designed for the light-tailed and bounded reward distributions arising in this setting and, thus, provides an efficient alternative compared to classic max k-armed bandit methods assuming heavy-tailed reward distributions. The authors theoretically and empirically evaluate this method on four standard AutoML benchmarks and demonstrate superior performance over prior approaches. They also release their code and data used in experiments.

**Questions:**

Please see the weaknesses above.

**Ethical Concerns:**

["NO or VERY MINOR ethics concerns only"]

**Final Justification:**

After reading the rebuttal and other reviewers' comments, I tend to weak accept.

**Limitations:**

Please see the weaknesses above.

**Quality:**

3

**Strengths And Weaknesses:**

Strengths:

1.	This paper proposes a state-of-the-art method for the CASH problem based on an extreme bandit algorithm MaxUCB. The authors provide regret bounds for MaxUCB and also justify their choice of exploration bonus for the type of distributions relevant to the CASH problem.
2.	The authors demonstrate the performance of the proposed method on four benchmarks, and corroborate that the proposed algorithm achieves good empirical performance on CASH.
3.	The authors release their code and data, which guarantees reproductivity.

Weaknesses:

1.	The regret definition in Eq. (4) is unusual in the bandit literature. The authors should discuss more on the motivation of this regret definition, combining with realistic hyperparameter optimization applications. Why should we consider the maximum over $t \leq T$?
2.	In hyperparameter optimization applications, an arm corresponds to a hyperparameter choice/setup. Then, in practical applications, the number of possible hyperparameter choices is large, which is often combinatorial with respect to the number of hyperparameters. Can the proposed method and theoretical guarantees be applied to the scenario when the number of possible hyperparameter choices (arms) is large?
3.	I feel that the proposed method and theoretical guarantees rely too much on the properties of the reward distributions in hyperparameter optimization tasks (what the authors discussed in the second paragraph of Section 3). But these properties may not always hold in realistic hyperparameter optimization applications.
4.	Theoretically, the proposed algorithm does not know the optimal choice of exploration parameter $\alpha$. So the regret bound in Corollary 4.3 may not be achieved by the proposed algorithm.
5.	The writing of this paper needs to be improved. Some definitions are missing, which makes some parts of this paper hard to understand. For example, what is $L_{i^*}$ in Theorem 4.2? What is two-level and single-level approaches for hyperparameter optimization mentioned in Section 5?

---

> ### Author Rebuttal · Authors · 2025-07-30
>
> Thanks a lot for the detailed review and the raised points.
>
> ---
> ### “1. The regret definition in Eq. (4) is unusual in the bandit literature.”
> We very much appreciate this question and are happy to clarify. Indeed, our regret definition (known as extreme regret [1]) is unusual in the general bandit literature, but it is central in HPO literature.
>
> Jamieson et al. (2015) [2] already state that max-value objectives should be the right ones for HPO, but rightfully state that they are “less well behaved”. Further, Nishihara et al. (2019) [3] stated that the **Max K-armed Bandit (Extreme Bandits) should be the valid framework for HPO** but also proved that without appropriate assumptions, it may be infeasible (lower bounds). More recently, Hu et al. (2021) [4] mentioned that the classic UCB algorithm is inappropriate for the CASH problem because of the regret definition. These very challenges motivated our study. A similar discussion can be found in lines 40-50, but we will extend it.
>
> ---
> ### “2. Can the proposed method and theoretical guarantees be applied to the scenario when the number of possible hyperparameter choices (arms) is large?”
> We apologize, but we are confused by this question. To clarify: In our setup, an arm is an ML model class, and “pulling an arm” corresponds to running HPO on the chosen  ML model class (see Figure 1 and lines 82-84). So, arms are **not** hyperparameter choices.
>
> MaxUCB would not be directly applicable to this different setup where an arm corresponds to a hyperparameter configuration (especially since our regret bound in Corollary 4.3 grows linearly with the number of arms K). Other Bandit methods, like Hyperband, are designed and known to work well for this different setting with only one model class.
>
> ---
> ### “3. But these properties may not always hold in realistic hyperparameter optimization applications.”
>
> Our data analysis covers several benchmarks, with 300+ datasets representing realistic hyperparameter optimization tasks (see response to Reviewer PkJH). Indeed, different problems may exist, but one important contribution of our paper is to consider the broader set of assumptions for the most realistic space of HPO/CASH problems.
>
> We derived theoretical guarantees under quite general semi-parametric assumptions, as stated in Theorem 4.2 and the accompanying discussion. Our assumptions allow for a wide range of reward distributions (we discuss typical distribution types in Appendix C.2). If a problem would have a very small L constant (in contrast to all the CASH problems we studied), then the performance of MaxUCB would be worse, as predicted by our theory.
>
> ---
> ### “4...So the regret bound in Corollary 4.3 may not be achieved by the proposed algorithm.”
>
> Yes, we completely agree. The result of Corollary 4.3 shows how to choose optimal $\alpha$ when these values are known or can be estimated in advance via meta-learning. In the discussion, we also mention that adapting $\alpha$ by estimating these parameters on the fly is possible.
>
> ---
> ### “5..The writing of this paper needs to be improved.”
> Thanks a lot for pointing this out. We will unify terminology in the revision:
>
> * $L_i$ refers to the value of $L$ from Lemma 3.3 for arm $i$, and $i^*$ denotes the optimal arm (see lines 224-225).
> * We refer to “combined search” as a single-level approach and with “two-level approach” to Bandit-based methods that address the bi-level optimization problem in Eq(2). See also our response to Reviewer 7Zgt.
>
> ---
> ###  References
> 1) Carpentier, Alexandra, and Michal Valko. "Extreme bandits." Advances in Neural Information Processing Systems 27 (2014).
> 2) Jamieson, Kevin, and Ameet Talwalkar. "Non-stochastic best arm identification and hyperparameter optimization." Artificial intelligence and statistics. PMLR, 2016.
> 3) Nishihara, Robert, David Lopez-Paz, and Léon Bottou. "No regret bound for extreme bandits." Artificial Intelligence and Statistics. PMLR, 2016.
> 4) Hu, Yi-Qi, et al. "Cascaded algorithm selection with extreme-region UCB bandit." IEEE Transactions on Pattern Analysis and Machine Intelligence 44.10 (2021): 6782-6794.

---

> > ### Comment · Reviewer_Vu3f · 2025-08-05
> > **I raised my rating from 3 to 4**
> >
> > Thanks for the response. My concerns were relieved. I raised my rating from 3 to 4.

---

### Official Review · Reviewer_7Zgt · 2025-07-02

**Clarity:** 4
**Significance:** 2
**Originality:** 2
**Rating:** 5
**Confidence:** 2

**Summary:**

This paper addresses the combined algorithm selection and hyperparameter optimization (CASH) problem using an extreme bandit framework with novel assumptions derived from empirical observations on AutoML benchmarks. Based on these assumptions, the authors propose a new extreme bandit algorithm, MaxUCB. The paper provides theoretical regret guarantees for the algorithm and offers guidance on selecting the exploration parameter alpha. Experimental results demonstrate state-of-the-art performance on several benchmark AutoML tasks.

**Questions:**

a)	The discussion in the introduction (lines 25–39) is somewhat confusing, especially the distinction between "combined algorithm" and "individual search-based algorithm," along with their respective pros and cons.

b)	What exactly is the difference between Eq. (1) and Eq. (2)? Are the optimization problems themselves different?

c)	Is the proposed algorithm completely agnostic to the inner HPO method?

d)	Typo in line 78: "trades off trade off"

e)	Regarding the earlier point on assumptions, while I appreciate the discussion in Section 3, my impression is that the effectiveness of algorithms for CASH will heavily depend on the task or data characteristics, as CASH is such a broad problem. What are the characteristics of the tasks and datasets used in the empirical analysis in Section 3? How do your observations vary across tasks or data types? More detailed analysis is needed here.

f)	Similarly, the authors mention that the choice of alpha may depend on the dataset characteristics. Can you elaborate on this? Is there any analysis or guidance on choosing alpha based on the task or data types?

g)	In Assumption 3.2, the random variable is assumed to be i.i.d., but line 86 suggests that the reward process is not i.i.d. Could you clarify?

h)	Typo in line 207: "property the maxima"

i)	In Section 5, I assume that the “two-level approach” and “single-level” correspond to “individual” and “combined” methods from the Introduction. Please unify the terminology throughout to avoid confusion, or explicitly define these terms.

j)	Typo in line 268: "study study"

**Ethical Concerns:**

["NO or VERY MINOR ethics concerns only"]

**Final Justification:**

I had rated this submission at accept. The authors provided some further clarifications to my questions, and as such at this point, I am leaving my score at accept.

**Limitations:**

Yes

**Quality:**

3

**Strengths And Weaknesses:**

2.	Strength
a)	The paper is clearly written and well-organized.
b)	It offers solid theoretical and algorithmic contributions.
c)	The limitations of the proposed method are explicitly discussed, along with appropriate cautions.

Weakness
a)	A more in-depth discussion of the assumptions based on empirical observations is needed. Since the algorithmic contribution relies heavily on these new assumptions and the resulting modification, further analysis would strengthen the work. Specific concerns are detailed in the Questions section below.
b)	From a purely algorithmic perspective, the originality of the proposed algorithm is somewhat unclear. It may be a minor modification of existing methods, though I acknowledge I am not an expert in AutoML or related bandit algorithms.

---

> ### Author Rebuttal · Authors · 2025-07-30
>
> Thanks a lot for the detailed review and acknowledging our contributions. We will fix all minor comments in a revised version and address your questions in the following.
>
> ---
> ### “From a purely algorithmic perspective, the originality of the proposed algorithm is somewhat unclear.”
>
> The UCB principle is central to dealing with decision-making under uncertainty. We agree it is not original in this sense. However, as Reviewer Vu3f noted, the regret definition in Eq. (4), which corresponds to the Max K-armed Bandit problem, differs from the common regret used in classical bandit settings. The main challenge lies in getting the details right in the modelling assumptions. Our main contribution, and what makes the method work ultimately, is to identify a semi-parametric model of the distribution of the maxima we aim to optimize. We discuss this in more detail in our response to Reviewer PkJH.
>
>
> ---
> ### “a) … distinction between "combined algorithm" and "individual search-based algorithm”
>
> With the “combined search” approach, we refer to an optimization method that searches the entire space of all K models’ hyperparameters, while an “individual search-based algorithm” only searches hyperparameters for a single model (and without budget allocation, needs to be conducted K times). The main challenge for “combined search” is efficiently searching **the entire high-dimensional and heterogeneous combination of K search spaces**. The main challenge for “individual search-based algorithms” is to **allocate the budget between the K runs** by an additional optimizer.
>
> As you correctly pointed out, the “two-level” and “single-level” terminology corresponds to the “individual” (i.e., decomposed CASH) and “combined search” approaches introduced earlier. We agree that terminology might be confusing, but we wanted to highlight the combined term in CASH; we will revise and unify the terminology.
>
> ---
> ### “b) difference between Eq. (1) and Eq. (2)?...”
> Eq.(1) and Eq.(2) are equivalent and describe the same optimization problem (and thus also share the same solution): finding the best-performing configuration of a model. In Eq. (2), we highlight the structure of CASH problems and that we can decompose the problem into a bi-level optimization problem.
>
> ---
> ### “c) agnostic to the inner HPO method ...”
> Yes, absolutely. It’s a strength of our method that it would also allow using a different HPO method (e.g., random search or Bayesian optimization) per model class.
>
>
> ---
> ### “e) … the characteristics of the tasks and datasets used in the empirical analysis in Section 3 ...”
>
> Yes, we agree. Tables D.1, D.2, and D.3 provide detailed information about the tasks, and Figures D.7 and D.8 report performance per task.  Each task corresponds to a different dataset. For YaHPOgym, the task is classification, with accuracy as the evaluation metric.
>
> For Reshuffling, the task is binary classification with AUC as the metric. In TabRepo and TabRepoRaw, the tasks include regression, multiclass classification, and binary classification, with RMSE, log loss, and AUC used as the respective metrics (we also discuss this in our response to reviewer PkJH)
>
> In general, linking MaxUCB's performance to task characteristics or using meta-learning to exploit similarities across tasks would be promising future directions. However, here, we first focused on designing and analysing a principled algorithm that works empirically well on average.
>
> ---
> ### “f) choice of alpha?”
>
> Corollary 4.3 theoretically gives the optimal value for $\alpha$ when characteristics of the datasets and the budget are known. We find that a default of $\alpha=0.5$ works empirically well (see Appendix D.4). Indeed, we can not provide a general rule for choosing the optimal alpha as it is a problem-dependent quantity. Similar to our answer to e), meta-learning the optimal alpha based on task characteristics could be an empirical approach to mitigate this issue.
>
> ---
> ### “g)  In Assumption 3.2, the random variable is assumed to be i.i.d., but line 86 suggests that the reward process is not i.i.d. Could you clarify?”
> You’re correct. In Assumption 3.2, we assume that our random variable is i.i.d. In practice, our rewards are not strictly i.i.d. (unless we use random search), and we know that this assumption might not hold in practice (as stated in line 86). However, our data analysis shows that the distribution shift during HPO is small in most cases (line 146), and our empirical results confirm that we can model this process reasonably well to perform well in practice. **This simplifying assumption is key to designing a simple and effective algorithm.**

---

### Decision · Program_Chairs · 2025-09-17

**Decision:**

Accept (poster)

**Comment:**

This work tackles the challenge of algorithm and hyperparameter selection in automated machine learning. The main contribution is an algorithm MaxUCB that can balance exploration of model classes with hyper-parameter optimization. The authors conduct a thorough evaluation of their experimental methods and derive regret bounds for their setting. The reviewers agreed that although the technical content behind the MaxUCB algorithm is not extremely technically challenge or novel, this work meets the bar for presentation at Neurips.